# ENaC-mediated sodium influx exacerbates NLRP3-dependent inflammation in cystic fibrosis

Thomas Scambler[1†], Heledd H Jarosz-Griffiths[1,2,3†], Samuel Lara-Reyna[1,2], Shelly Pathak[1], Chi Wong[1,2,3], Jonathan Holbrook[1,2,3], Fabio Martinon[3,4], Sinisa Savic[1,3,5], Daniel Peckham[2,3,6‡], Michael F McDermott[1,3‡*]

[1]Leeds Institute of Rheumatic and Musculoskeletal Medicine, University of Leeds, Leeds, United Kingdom; [2]Leeds Institute of Medical Research, University of Leeds, Leeds, United Kingdom; [3]Leeds Cystic Fibrosis Trust Strategic Research Centre, University of Leeds, Leeds, United Kingdom; [4]Department of Biochemistry, University of Lausanne, Lausanne, Switzerland; [5]Department of Clinical Immunology and Allergy, St James's University Hospital, Leeds, United Kingdom; [6]Adult Cystic Fibrosis Unit, St James' University Hospital, Leeds, United Kingdom

*For correspondence:
M.McDermott@leeds.ac.uk

†These authors contributed equally to this work
‡These authors also contributed equally to this work

Competing interests: The authors declare that no competing interests exist.

**Abstract** Cystic Fibrosis (CF) is a monogenic disease caused by mutations in the cystic fibrosis transmembrane conductance regulator (CFTR) gene, resulting in defective CFTR-mediated chloride and bicarbonate transport, with dysregulation of epithelial sodium channels (ENaC). These changes alter fluid and electrolyte homeostasis and result in an exaggerated proinflammatory response driven, in part, by infection. We tested the hypothesis that NLRP3 inflammasome activation and ENaC upregulation drives exaggerated innate-immune responses in this multisystem disease. We identify an enhanced proinflammatory signature, as evidenced by increased levels of IL-18, IL-1β, caspase-1 activity and ASC-speck release in monocytes, epithelia and serum with CF-associated mutations; these differences were reversed by pretreatment with NLRP3 inflammasome inhibitors and notably, inhibition of amiloride-sensitive sodium ($Na^+$) channels. Overexpression of β-ENaC, in the absence of CFTR dysfunction, increased NLRP3-mediated inflammation, indicating that dysregulated, ENaC-dependent signalling may drive exaggerated inflammatory responses in CF. These data support a role for sodium in modulating NLRP3 inflammasome activation.
DOI: https://doi.org/10.7554/eLife.49248.001

## Introduction

Cystic fibrosis (CF) is the most common life-threatening autosomal recessive disease to affect Caucasian populations. Mutations in the cystic fibrosis transmembrane conductance regulator (CFTR) result in reduced expression and function of the CFTR with the most common mutation (ΔF508/ΔF508) resulting in inadequate processing of the protein and subsequent intracellular trapping in the endoplasmic reticulum (ER) (*Elborn, 2016*). Clinical manifestations of this debilitating condition include repeated pulmonary infections, innate immune-driven episodes of inflammation and inflammatory arthritis (*Elborn, 2016*; *Whitsett and Alenghat, 2015*; *Bals et al., 1999*; *Montgomery et al., 2017*). The CFTR protein is widely expressed in a variety of cells and tissues where it functions as an anion channel, conducting chloride ($Cl^-$) and bicarbonate (HCO3-) ions, and as a regulator of a range of epithelial transport proteins, including the epithelial sodium channel (ENaC) (*König et al., 2002*; *Konstas et al., 2003*; *Kunzelmann, 2003*; *Berdiev et al., 2009*).

The NLRP3 inflammasome is an important inflammatory pathway in CF but there is little indication as to whether this reflects underlying innate autoinflammation or activation in response to chronic

bacterial, viral and fungal infections, including pathogens such as *Pseudomonas aeruginosa,* and *Burhkholderia cepacia complex* (*Iannitti et al., 2016*; *Rimessi et al., 2015*; *Montgomery et al., 2017*; *Fritzsching et al., 2015*; *Kiedrowski and Bomberger, 2018*). We propose that CF exhibits many hallmarks of an autoinflammatory condition (*Peckham et al., 2017*; *McGonagle and McDermott, 2006*; *McDermott et al., 1999*), with infiltration by innate immune cells (macrophages and neutrophils) at target sites, and a lack of autoantibodies or autoreactive T cells (*McDermott et al., 1999*). The NLRP3 intracellular protein complex is a sensor that detects changes in cellular homeostasis rather than directly sensing common pathogenic or endogenous motifs. Multiple cellular events have been observed to trigger NLRP3 activation, including $K^+$ efflux, $Na^+$ influx, $Cl^-$ efflux and $Ca^{2+}$ signalling (*Schorn et al., 2011*; *Muñoz-Planillo et al., 2013*; *Katsnelson and George, 2013*; *Domingo-Fernández et al., 2017*; *Green et al., 2018*; *Hafner-Bratkovič and Pelegrín, 2018*). All known canonical inflammasomes, including the NLRP3 inflammasome, function as signalling platforms for caspase-1-driven activation of IL-1-type cytokines (IL-1β and IL-18). One of IL-18's main functions is induction of cell-mediated immunity and IFN-γ secretion by natural killer (NK) and T-cells, whereas IL-1β has a central role in inducing fever with immune cell proliferation, differentiation and apoptosis. Inflammasome activation induces a programmed cell death downstream of the activation of caspases-1, 4, 5, and 11. Gasdermin D (GSDMD) is cleaved by said caspases, oligomerizing and forming pores (13 nm) in the plasma membrane, releasing mature IL-1β and IL-18 and triggering cell lysis, or pyroptosis (*Cookson and Brennan, 2001*; *Platnich and Muruve, 2019*).

In healthy lungs, ENaC helps maintain normal volume and composition of airway surface liquid (ASL). An absence or reduction in functional CFTR leads to defective CFTR-mediated anion transport and upregulation of ENaC. These changes in normal homeostasis result in fluid hyperabsorption, abnormally thick viscous mucus and defective mucociliary clearance (*Althaus, 2013*; *Boucher, 2019*). While there is no literature to support a direct link between ENaC and inflammation in CF, there is indirect evidence to suggest that aberrant sodium ($Na^+$) transport influences the disease process. Overexpression of β-ENaC in mice, results in CF-like lung disease, with ASL dehydration, inflammation and mucous obstruction of bronchial airways (*Mall et al., 2004*; *Zhou et al., 2011*; *Zhou et al., 2008*). In humans, genetic variants in the *β*- and *γ*- ENaC chains, leading to functional abnormalities in ENaC, have been associated with bronchiectasis and CF-like symptoms (*Fajac et al., 2008*). By contrast, rare mutations associated with hypomorphic ENaC activity, can slow disease progression in patients homozygous for the CFTR ΔF508 mutation (*Mall et al., 2004*; *Zhou et al., 2011*; *Donaldson and Boucher, 2007*). These studies highlight the essential role of ENaC in regulating normal airway homeostasis and show that inhibition of ENaC may modify disease progression, either by altering ASL composition or modifying other processes, such as inflammation. Several trials are ongoing to assess the safety and effectiveness of new topical ENaC inhibitors to restoring airway surface liquid and mucociliary clearance in CF (*Cystic Fibrosis Foundation, 2017*).

The aims of this study were to characterise NLRP3 inflammasome activation in CF and to investigate the role of the epithelial $Na^+$ channel, ENaC, in driving this inflammation through alterations in ionic homeostasis, a known NLRP3 activating event.

In order to fulfil these aims, monocytes and epithelial cells with characterised CF-associated mutations are directly compared to cohorts of NCFB and SAID. The NCFB cohort comprises of individuals with primary ciliary dyskinesia (PCD), a rare, ciliopathic, autosomal recessive genetic disorder affects the movement of cilia in the lining of the respiratory tract. Individuals with PCD suffer from reduced mucus clearance from the lungs, and susceptibility to chronic recurrent respiratory infections, as is the case with CF. By comparing monocytic- and epithelial-driven inflammation in CF and PCD, one is able to distinguish between inflammation due to recurrent infection, as is the case with both CF and NCFB, and inflammation that is downstream of CFTR/ENaC-mediated ionic disturbances, specific to CF.

The SAID patient cohort is composed of an array of systemic autoinflammatory diseases that are defined by an innate immune driven inflammation. The variety of autoinflammatory disorders described in this manuscript demonstrates the broad range of pathophysiology within this rare inflammatory disease spectrum. Here we demonstrate that the intrinsic ionic defect in cells and individuals with CF-associated mutations predisposes hyperactivation of the NLRP3 inflammasome, leading to inappropriate and destructive innate immune driven inflammation, as found in autoinflammation.

## Results

### Increased NLRP3-dependent IL-18 secretion in human bronchial epithelial cells with CF-associated mutations

In CF, airway epithelial cells have been shown to produce exaggerated levels of proinflammatory cytokines (IL-8 and TNF) characteristic of a hyper-inflammatory phenotype (*Venkatakrishnan et al., 2000*). However, IL-1β secretion is barely detectable in bronchial/airway epithelial cells and does not greatly increase following stimulus with NLRP3-inflammasome activators, despite being highly responsive to the effects of the IL-1β cytokine itself (*Tang et al., 2012*; *Peeters et al., 2013*; *Gillette et al., 2013*). We explored the effects of the constitutively expressed IL-18 cytokine in HBECs (BEAS-2B (WT) control, IB3-1 (ΔF508/W1282X), CuFi-1 (ΔF508/ΔF508) and CuFi-4 (ΔF508/ G551D)) following treatment with activators for the four main caspase-1 driven inflammasomes, NLRC4, pyrin, AIM2 and NLRP3 (*Figure 1*).

Under basal conditions, IL-18 levels in the HBECs were undetectable in the absence of a stimulus. All inflammasomes were primed with lipopolysaccharide from *Escherichia coli* K12 (LPS), which specifically targets TLR4 and is used to promote *pro-IL-18/IL-1β* expression; this was followed by stimulation with established activators of the inflammasomes, NLRC4 (flagellin), pyrin (TcdB) and AIM-2 (dsDNA). There were no differences between cells with or without CF-associated mutations for activation of these particular inflammasomes (*Figure 1A*). However, when HBECs were stimulated with LPS, followed by ATP, a specific NLRP3 inflammasome activating signal that nucleates NLRP3 assembly with ASC, pyrin and caspase-1, IL-18 secretion was upregulated in CF-associated mutant cell lines, IB3-1 (p<0.0001) and CuFi-1 (p<0.0001) relative to the BEAS-2B control. When these cells were pre-treated with small molecule inhibitors of the NLRP3 inflammasome signalling pathway (MCC950; NLRP3, OxPAPC;TLR4, YVAD;caspases), IL-18 secretion was reduced (*Figure 1B*) in the CF-associated mutant HBEC lines thereby confirming NLRP3 inflammasome as the source of the elevated IL-18 inflammatory cytokine. Consistent with increased IL-18 cytokine levels, caspase-1 activation was also elevated in the CF-associated mutant HBEC lines relative to control post-LPS and ATP stimulation in vitro, and was depleted by MCC950 pre-treatment (*Figure 1C*).

It is well established that NLRP3 inflammasome triggers sterile inflammatory responses and pyroptosis, which is a proinflammatory form of programmed cell death initiated by the activation of inflammatory caspases (*Bergsbaken et al., 2009*). To examine this, we monitored cell death whereby pyroptosis was distinguished from necrosis by pretreating cells with a caspase inhibitor and using lactose dehydrogenase (LDH) as a measure of necrosis. Elevated pyroptosis was present after LPS and ATP stimulation in the HBEC line IB3-1 (*Figure 1D*), consistent with an NLRP3-mediated hyper-inflammatory phenotype.

### Increased NLRP3-dependent IL-1β/IL-18 secretion in human monocytes with CF-associated mutations

We next explored NLRP3 inflammasome activation in primary monocytes (main producers of IL-18 and IL-1β, along with neutrophils) derived from HC, CF, SAID, and NCFB (*Figure 2*). Under basal conditions primary monocytes, isolated from HC and CF, showed no significant difference in the secretion of IL-18 and IL-1β cytokines (*Figure 2A,B*) or when monocytes were stimulated with LPS alone across all patient groups (*Figure 2—figure supplement 1A,B*). As with the HBECs, there was also no statistical difference between HC and CF in LPS mediated activation of inflammasomes, NLRC4, pyrin and AIM-2.

In primary monocytes stimulated with LPS, followed by ATP to activate the NLRP3 inflammasome, we observed hyper-responsiveness in IL-18 (p<0.0001) and IL-1β (p=0.0009) secretion in CF monocytes relative to HC. When these cells were pretreated with NLRP3 inflammasome pathway inhibitors (MCC950; NLRP3, OxPAPC;TLR4, YVAD;caspase-1), their secretions were significantly abrogated across all patient groups (HC, CF, SAID and NCFB) (*Figure 2C,D*). Downstream, IL-18 acts on NK and T-cells to express and secrete IFN-γ (*Kim et al., 2015*); we monitored *IFN-γ* gene expression and secretion and found they were increased, post-NLRP3-inflammasome activation, in PBMCs from patients with CF compared to HC (*Figure 2—figure supplement 1C,D*).

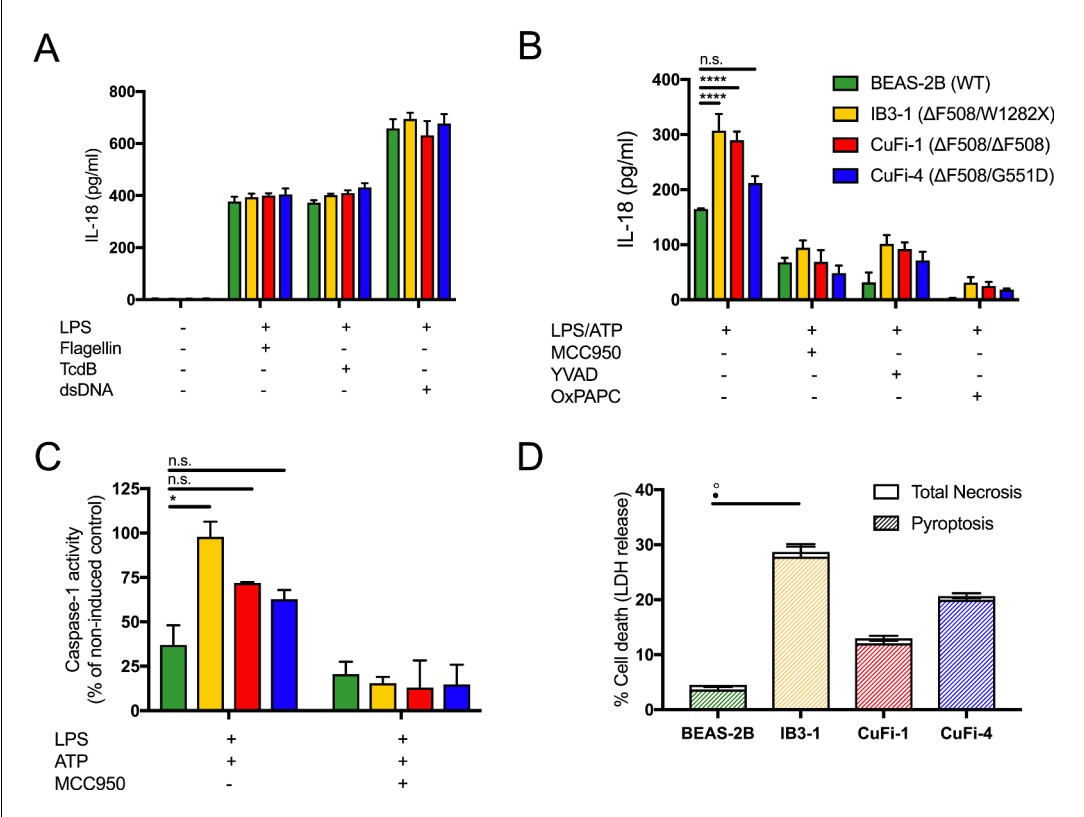

**Figure 1.** LPS-induced IL-18 secretion in human bronchial epithelial cells is higher in cells with CF-associated mutations and is NLRP3 inflammasome dependent. Human bronchial epithelial cell (HBEC) lines (BEAS-2B (WT), IB3-1 (ΔF508/W1282X), CuFi (ΔF508/ ΔF508), CuFi4 (ΔF508/G551D) (n = 3 independent experiments) were unstimulated or stimulated with Lipopolysaccharide, from *Escherichia coli K12* (LPS Ultrapure), which specifically targets TLR4 (10 ng/mL) for 4 hr before being stimulated for 4 hr with Flagellin (10 ng/mL with Lipofectamine 2000) for NLRC4 inflammasome, TcdB (10 ng/mL) for Pyrin inflammasome or poly(dA:dT) dsDNA (1 µg/mL with Lipofectamine 2000) for AIM2 inflammasome. ELISA assays were used to detect (A) IL-18. To monitor NLRP3 inflammasome activation, HBEC (n = 3 independent experiments) were pre-incubated with MCC950 (15 µM), OxPAPC (30 µg/mL) and YVAD (2 µg/mL) for 1 hr before a stimulation with LPS (10 ng/mL, 4 hr), and ATP (5 mM) for the final 30 min. ELISA assays were used to detect (B) IL-18 and (D) colourimetric assay used to detect caspase-1 activity in protein lysates for LPS/ATP and LPS/ATP/MCC950). (D) Necrosis and pyroptosis are represented as superimposed bar charts. Total necrosis was measured using LDH release assay. For pyroptotic cell death, each sample/condition was repeated in parallel with a caspase-1 inhibitor (YVAD (2 mg/mL, 1 hr)) pre-treatment. The total necrosis level was then taken away from the caspase-1 inhibited sample, or 'caspase-1 independent' necrosis, with the remaining LDH level termed 'caspase-1 dependent necrosis' or pyroptosis. Cells were then stimulated with LPS (10 ng/mL, 4 hr), and ATP (5 mM) for final 30 min. The assay was performed with HBEC lines (n = 3 independent experiments). (○) Significance for Total Necrosis (●) Significance for Pyroptosis. A 2-way ANOVA with Tukey's multiple comparison test was performed (p values * =< 0.05, ** =< 0.01, *** =< 0.001 and **** =< 0.0001).

DOI: https://doi.org/10.7554/eLife.49248.002

We next examined cell death in HC, CF, SAID and NCFB monocytes (*Figure 2G*). Elevated pyroptosis was present after LPS and ATP stimulation in monocytes from patients with CF, and also in those diagnosed with SAID. Notably, caspase-1-independent necrosis was also elevated in CF and SAID monocytes (*Figure 2E*). To understand the relationship between elevated pyroptosis and downstream inflammation, the presence of ASC protein aggregates (specks), key inflammasome components, were measured in the cell supernatants, post-NLRP3 inflammasome activation. ASC-specks were elevated in stimulated monocytes from patients with CF-associated mutations, and also in those diagnosed with SAID (*Figure 2F*), and were reduced with MCC950 treatment. Similar to HBECs, caspase-1 activation was also elevated in CF and SAID monocytes, post-LPS and ATP stimulation in vitro, and were depleted by MCC950 pretreatment in the monocytes (*Figure 2E*).

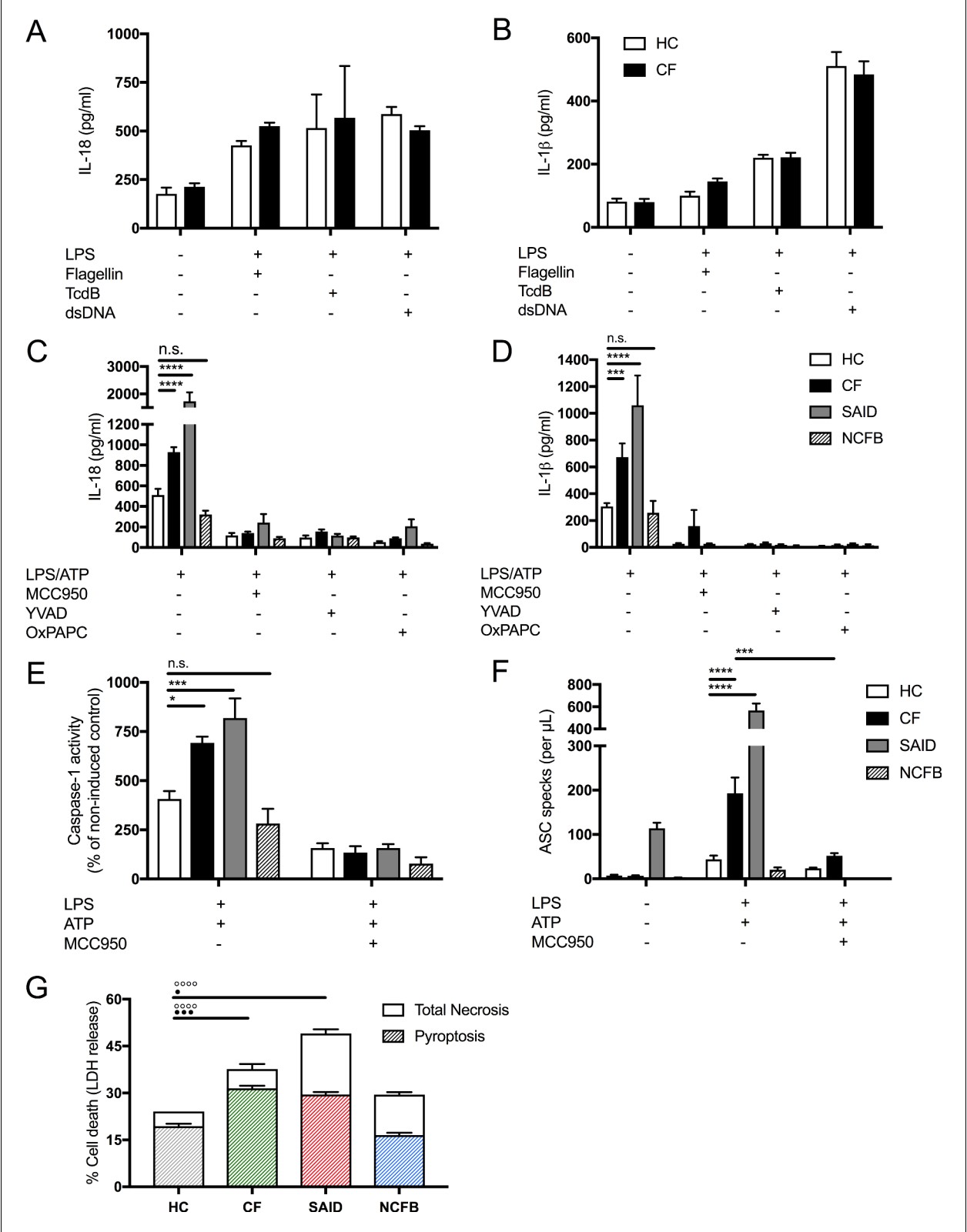

**Figure 2.** LPS-induced IL-1β/IL-18 secretion in human monocytes is higher in CF and is NLRP3 inflammasome dependent. Primary monocytes from HC and CF (HC, n = 10; CF, n = 10) were unstimulated or stimulated with LPS which specifically targets TLR4 (10 ng/mL) for 4 hr before being stimulated for 4 hr with Flagellin (10 ng/mL with Lipofectamine 2000) for NLRC4 inflammasome, or TcdB (10 ng/mL) for Pyrin inflammasome or poly(dA:dT) dsDNA (1 μg/mL with Lipofectamine 2000) for AIM2 inflammasome. ELISA assays were used to detect (**A**) IL-18 and (**B**) IL-1β cytokine secretion in supernatants. To

*Figure 2 continued on next page*

*Figure 2 continued*

monitor NLRP3 inflammasome activation, primary monocytes from HC, CF, SAID and NCFB (HC, n = 10; CF, n = 10; SAID, n = 4; NCFB, n = 4) were pre-incubated with MCC950 (15 µM), OxPAPC (30 µg/mL) and YVAD (2 µg/mL) for 1 hr before a stimulation with LPS (10 ng/mL, 4 hr), and ATP (5 mM) for the final 30 min. ELISA assays were used to detect (C) IL-18 and (D) IL-1β cytokine secretion in supernatants and (E) a colourimetric assay was used to detect caspase-1 activity in protein lysates (HC, n = 10; CF, n = 10; SAID, n = 4; NCFB, n = 4). (F) Flow cytometry was used to detect ASC specks in supernatants of primary monocytes from HC, CF, SAID and NCFB (HC, n = 10; CF, n = 10; SAID, n = 6; NCFB, n = 4) for ±LPS/ATP and (HC, n = 5; CF, n = 5) for MCC950 with LPS/ATP. (G) Necrosis and pyroptosis are represented as superimposed bar charts. Total necrosis was measured using LDH release assay. For pyroptotic cell death, each sample/condition was repeated in parallel with a caspase-1 inhibitor (YVAD (2 mg/mL, 1 hr)) pre-treatment. The total necrosis level was taken away from the caspase-1 inhibited sample, or 'caspase-1 independent' necrosis, with the remaining LDH level termed 'caspase-1 dependent necrosis' or pyroptosis. Cells were then stimulated with LPS (10 ng/mL, 4 hr), and ATP (5 mM) for final 30 min. The assay was performed with primary monocytes from HC, CF, SAID and NCFB (HC, n = 10; CF, n = 10; SAID, n = 4; NCFB, n = 4). (○) Significance for Total Necrosis (●) Significance for pyroptosis. A 2-way ANOVA statistical test was performed, with Tukey post-hoc correction (p values * =< 0.05, ** =< 0.01, *** =< 0.001 and **** =< 0.0001; *error bars* ± SEM). Inhibitor treatments in panels a-c were found to significantly reduce cytokine secretion and caspase-1 activity to **p =< 0.01 or less, for CF and SAID groups respectively. Significance values not displayed on the graph.

DOI: https://doi.org/10.7554/eLife.49248.003

The following figure supplement is available for figure 2:

**Figure supplement 1.** Primary monocytes from HC, CF, SAID and NCFB (HC n = 9, CF n = 9, SAID n = 4, NCFB n = 4) were unstimulated or stimulated with LPS (10 ng/ml, 4 hr) or LPS (10 ng/ml, 4 hr) and ATP (5 mM) for the final 30 min.

DOI: https://doi.org/10.7554/eLife.49248.004

## Proinflammatory cytokines and ASC specks are elevated in CF Sera, and are comparable to patients diagnosed with systemic autoinflammatory disease (SAID)

To understand the extent of systemic inflammation in CF, serum cytokine levels were measured in patients with CF, SAID, NCFB and HC. Serum IL-18 (p=0.0064), IL-1β (p<0.0001) and IL-1Ra (p<0.0001) levels were all significantly elevated in CF samples, with levels comparable to SAIDs (*Figure 3A–C*). However, in contrast to IL-1-type cytokines, the inflammasome-independent cytokines, TNF and IL-6, were not significantly elevated in patients with CF whereas the levels of these two cytokines were significantly elevated in samples from patients with SAID (*Figure 3—figure supplement 1*). All SAID patients were on active recombinant IL-1Ra (anakinra) therapy, which will have reduced serum cytokine levels. However, levels of IL-18 and endogenous IL-1Ra were raised in patients with SAID, and are comparable to the proinflammatory IL-1 cytokines found in CF-serum.

We next sought to confirm that serum IL-1-type cytokines were associated with NLRP3 inflammasome activation by detecting the presence of ASC specks, in sera. ASC-specks were significantly elevated in CF (p=0.0007) and SAID sera compared to HC (*Figure 3D*), reflecting inflammasome-mediated inflammation and pyroptosis (*Franklin et al., 2014*). The activity of caspase-1, the rate-limiting factor in the activation of all inflammasomes (*Mariathasan et al., 2006*), was significantly elevated in CF and SAID serum samples compared to NCFB and HC (*Figure 3E*).

These data suggest that systemic serum cytokine levels from patients with CF are comparable to patients diagnosed with SAID and can be characterised by release of proinflammatory IL-1-type cytokine family members (IL-1β and IL-18), associated with NLRP3 inflammasome activation.

## Dysregulated na⁺ and K⁺ in cells with CF-associated mutations can be modulated with ENaC inhibitors

As K⁺ efflux and Na⁺ influx are thought to occur upstream of NLRP3 inflammasome activation and ENaC overactivation is a recognised event in cells with CF-associated mutations, we monitored intracellular concentrations of K⁺ and Na⁺ to determine if they were dysregulated in CF. The concentration gradient of Na⁺ and K⁺ is essential for cellular homeostasis, including resting potential, nutrient transport, cell volume and signal transduction. Here we tested the hypothesis that dysregulated ENaC-mediated Na⁺ transport alters the Na⁺/K⁺ gradient, enhancing K⁺ efflux and downstream NLRP3 inflammasome activation. We found that intracellular Na⁺ levels were significantly higher in CF cells [monocytes (p<0.0001), HBEC (p<0.0001)], at the peak of Na⁺ influx, which was recorded immediately following stimulus with ATP. This corresponded with a superior reduction in intracellular K⁺ [monocytes (p<0.0001), HBEC (p<0.0001)], suggestive of greater K⁺ efflux in these cells, upon ATP stimulation (*Figure 4A,B,E,F*, *Figure 4—figure supplement 1*). The magnitude of the observed

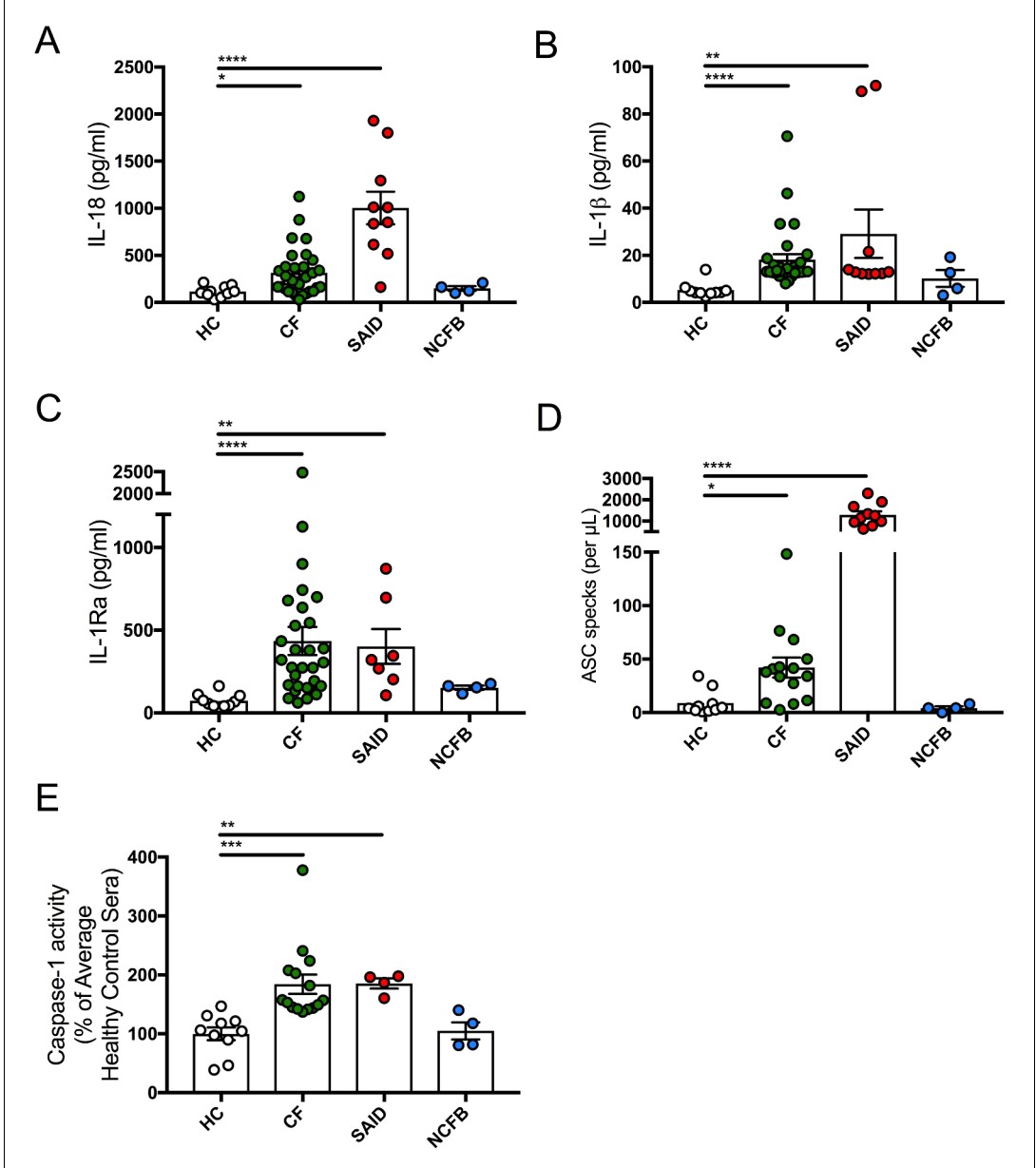

**Figure 3.** Inflammatory serum cytokine signature in CF. (**A**) ELISA assays were used to detect IL-18 (HC, n = 10, CF, n = 30, SAID, n = 10, NCFB, n = 4), (**B**) IL-1β (HC, n = 10, CF, n = 30, SAID, n = 10, NCFB, n = 4), (**C**) IL-1Ra (HC, n = 10, CF, n = 30, SAID, n = 7, NCFB, n = 4) in patient sera. Outliers in SAID group for IL-1β and IL-1Ra correspond to HIDS one and A20 deficiency (**D**) Flow cytometry was used to detect ASC specks (HC, n = 10, CF, n = 15, SAID, n = 10, NCFB, n = 4) in patient sera. (**E**) A colorimetric assay to detect caspase-1 activity in sera of patients with CF, SAID and NCFB as a percentage of HC (HC, n = 10, CF, n = 15, SAID, n = 4, NCFB, n = 4). Of note, an undetermined amount of detected IL-1Ra is attributed to circulating Anakinra (recombinant IL-1Ra) specifically in the SAID cohort. The Kruskal-Wallis non-parametric test, with Dunn's multiple comparison test, was performed (p values * =< 0.05, ** =< 0.01, *** =< 0.001 and **** =< 0.0001; *error bars* ± S.E.M).

DOI: https://doi.org/10.7554/eLife.49248.005

The following source data and figure supplements are available for figure 3:

**Source data 1.** Serum cytokine levels for all patient groups.
DOI: https://doi.org/10.7554/eLife.49248.008

**Figure supplement 1.** Inflammatory serum cytokine signature in CF.
DOI: https://doi.org/10.7554/eLife.49248.006

**Figure supplement 1—source data 1.** Serum cytokine levels for all patient groups.
DOI: https://doi.org/10.7554/eLife.49248.007

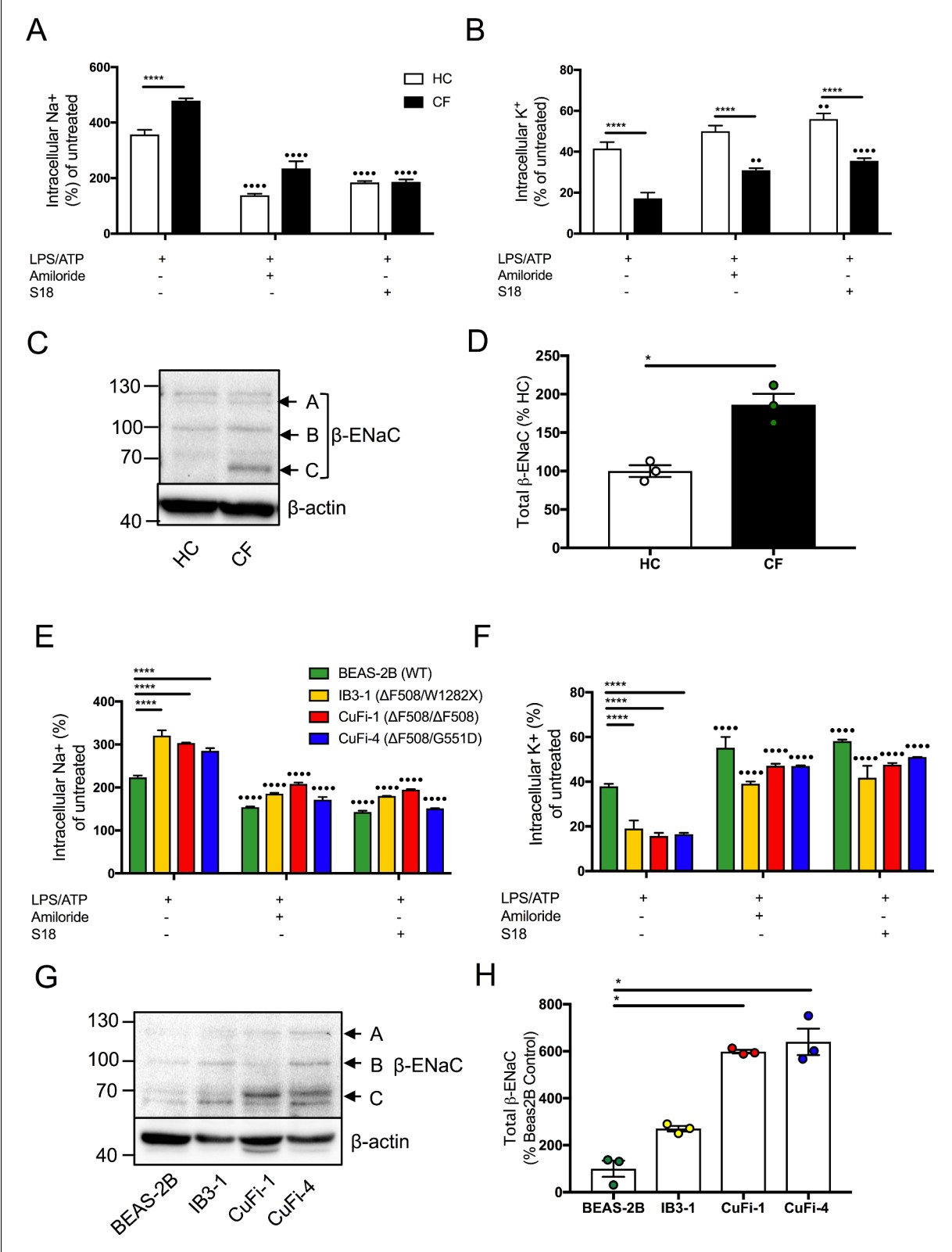

**Figure 4.** Dysregulated Na$^+$ and K$^+$ in cells with CF-associated mutations can be modulated with ENaC inhibitors. Intracellular Na$^+$ was detected using an AM ester of sodium indictor SBFI (S-1263) and (**B, D**) intracellular K$^+$ was detected using an AM ester of potassium indictor PBFI (P-1266); changes in fluorescence were measured by fluorimeter post-stimulation with 5 mM ATP in (**A, B**) monocytes (HC = 7, CF = 7) (**E, F**) HBECs (n = 3 independent experiments). Cells were pre-treated with the following: amiloride (100 μM), S18 derived peptide (25 μM, 4 hr) with LPS (10 ng/mL, 4 hr) and ATP (5 mM)
*Figure 4 continued on next page*

*Figure 4 continued*

for the final 30 min. A 2-way ANOVA with Tukey's multiple comparison test was performed (p values * =< 0.05, ** =< 0.01, *** =< 0.001 and **** =< 0.0001) (*) indicate significance when comparing HC with CF-associated mutants. (•) indicate significance between treatments within the same cell line. (C) Endogenous β-ENaC protein expression was detected using western blot in BEAS-2B HBEC, HC and CF monocytes (C) and densitometry analysis of total β-ENaC (bands A, B, C indicated on blot) was quantified in (D) for CF relative to HC (n = 3 independent experiments). (G) BEAS-2B, IB3-1, CuFi-1 and CuFi-4 HBEC lines and densitometry analysis of total β−ENaC (bands A, B, C indicated on blot) was quantified in (H) (n = 3 independent experiments). Band A represents complex N-Glycosylation, 110 kDa β-ENaC (found when associated as ENaC complex); Band B represents Endo-H sensitive N-Glycosylation, 96 kDa β-ENaC; Band C represents immature non-glycosylated, 66 kDa β−ENaC. The Mann-Whitney non-parametric test was performed (p values * =≤ 0.05).

DOI: https://doi.org/10.7554/eLife.49248.009

The following figure supplement is available for figure 4:

**Figure supplement 1.** Increased sodium influx in cells with CF-associated mutations.
DOI: https://doi.org/10.7554/eLife.49248.010

Na$^+$ and K$^+$ fluxes was significantly reduced by ENaC inhibition (using both amiloride and the SPLUNC1-derived peptide, S18, a highly stable and specific small molecule inhibitor of ENaC channels). Pretreatment with 5-(N-ethyl-N-isopropyl)-amiloride, EIPA, a broad-spectrum Na$^+$ channel inhibitor with lower potency for ENaC, also reduced intracellular Na$^+$ and increased intracellular K$^+$ in CF monocytes and HBEC lines, but not to the same extent as amiloride or S18 (*Figure 4A,B,E,F*, *Figure 4—figure supplement 1*). To corroborate our findings, we used ouabain, a Na$^+$/K$^+$-ATPase inhibitor, which was associated with a higher intracellular Na$^+$ in CF cells compared to HC. Collectively, these data suggest that higher intracellular Na$^+$ levels in CF, mainly driven by ENaC, predispose cells to a greater K$^+$ efflux upon stimulation with ATP. Na$^+$ influx has been described as a modulator of NLRP3 activation, dependent on K$^+$ efflux (*Schorn et al., 2011*; *Muñoz-Planillo et al., 2013*; *Katsnelson and George, 2013*).

As inhibition of ENaC activity (by amiloride and S18) modulated elevated intracellular Na$^+$ levels in cells with CF-associated mutations, we measured β-ENaC protein expression in HBEC lines and found significantly increased expression, in CuFi-1 and CuFi-4 cells relative to BEAS-2B control (p=<0.05) (*Figure 4G,H*). Furthermore, the *β*-ENaC gene was expressed in monocytes with a significant increase in β-ENaC protein levels noted in CF monocytes compared to HC (p=<0.05) (*Figure 4—figure supplement 1*, *Figure 4C,D*).

## Inhibition of amiloride-sensitive sodium channels modulates inflammation in CF

We next sought to determine the extent to which dysregulated Na$^+$ levels contribute to the observed NLRP3 inflammasome activation in cells with CF-associated mutations. We utilised small molecule inhibitors, for potent inhibition of ENaC, for in vitro NLRP3 inflammasome activation assays. Notably, amiloride alleviated the augmented cytokine secretion, as well as caspase-1 activity in both primary CF monocytes (IL-18 p=0.0001; IL-1β p=0.0272) and HBEC lines (*Figure 5A,B,E*) (*Montgomery et al., 2017*; *Fritzsching et al., 2015*; *Mall et al., 2004*). S18 potently inhibited cytokine secretion (IL-18 p=0.0052; IL-1β p=0.0100) and caspase-1 activity exclusively in cells with CF-associated mutations (*Figure 5C*, *Figure 5—figure supplement 1*), whereas EIPA had no effect (*Figure 5—figure supplement 1A,C,D,E*). These data were replicated by nigericin activation of NLRP3, used as a control for ATP, due to ATP's ability to modulate other ionic channels (*Figure 5—figure supplement 2*). To corroborate these findings, both amiloride (p<0.0001) and S18 (p=0003) inhibited ASC-speck formation in CF monocytes (*Figure 5D*), and finally, inhibition of amiloride-sensitive channels did not modulate TNF levels (*Figure 5—figure supplement 1F*), suggestive of a specific ENaC-NLRP3 axis.

## β-ENaC overexpression in BEAS-2B cells increases proinflammatory cytokine secretion

In order to recapitulate the ENaC-NLRP3 axis, as revealed in this study, we overexpressed the β-ENaC chain in the WT BEAS-2B line. This approach has been used previously in a β-ENaC Tg-mouse model of CF, which recreated a CF-like lung disease state, with mucous plugging and excessive inflammation (*Mall et al., 2004*; *Zhou et al., 2011*). Overexpression of β-ENaC induced elevated IL-

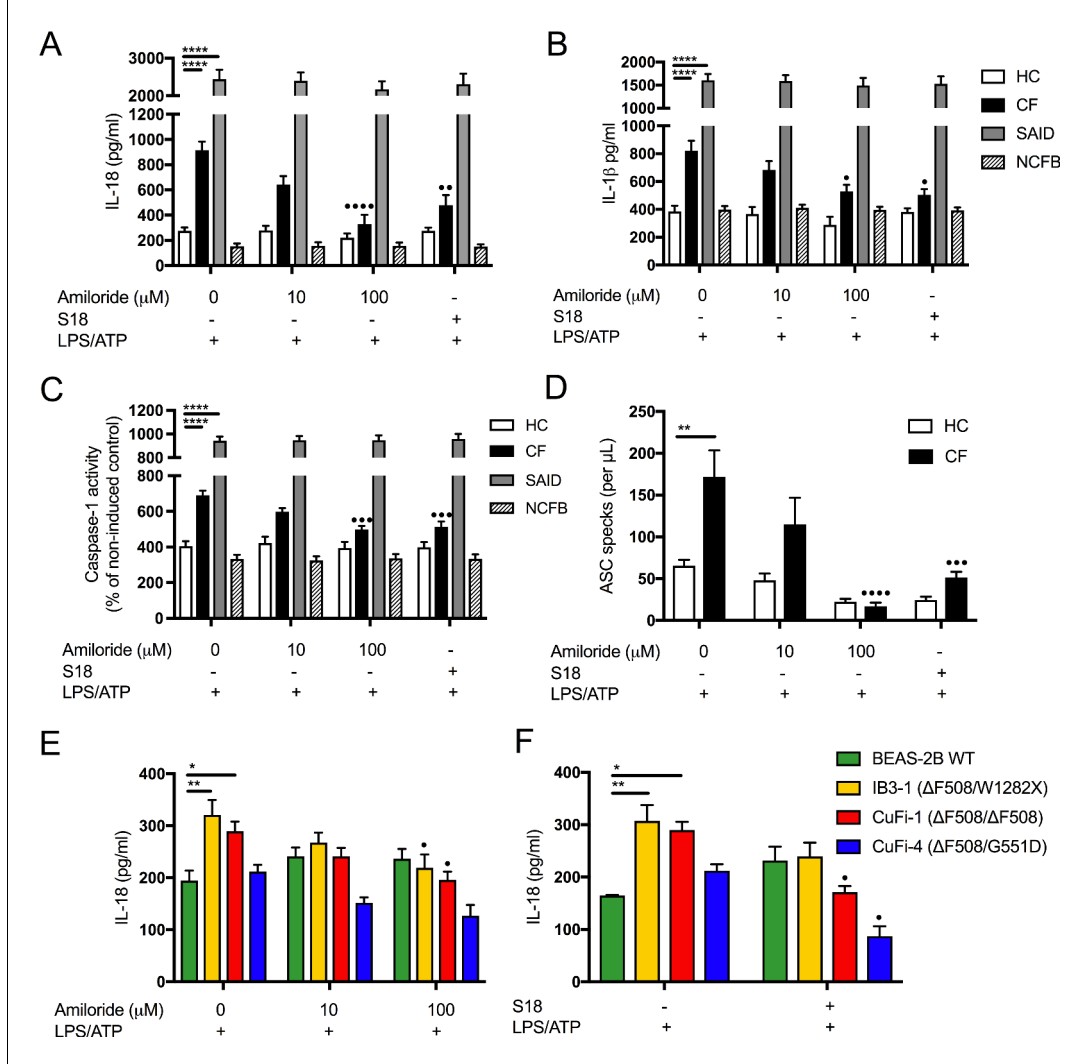

**Figure 5.** Inhibition of amiloride-sensitive sodium channels modulates inflammation in cells with CF-associated mutations. ELISA assays were used to detect IL-18 (A) and (B) IL-1β in monocytes from HC (n = 9 amiloride, n = 10 S18), patients with CF (n = 10), SAID (n = 4) and NCFB (n = 4) and IL-18 (E) HBEC (n = 3, amiloride independent experiments) (F) HBEC (n = 3, S18 independent experiments). (C) Colourimetric assay was used to detect caspase-1 activity in protein lysates (HC n = 11, CF n = 11) and (D) flow cytometry was used to detect ASC specks in supernatant of primary monocytes (HC n = 5, CF n = 5). Cell stimulation was as follows: Amiloride (100 μM or 10 μM, 1 hr) or S18 derived peptide (25 μM, 4 hr) were used as a pre-treatment before a stimulation with LPS (10 ng/mL, 4 hr) and ATP (5 mM) for the final 30 min. (E, F) SCNN1B over-expression in BEAS-2B cells increases pro-inflammatory cytokine secretion. (E) BEAS-2B cells were transiently transfected with 10μg SCNN1B cDNA (+) or a pcDNA3.1 vector only control (-) for 48 hr then stimulated with LPS (10 ng/mL, 4 hr) and ATP (5 mM) for the final 30 min (n = 3 independent experiments). Cells were lysed and immunoblotted for β-ENaC and β-actin. (F) ELISA assays were used to detect IL-18 in the supernatant fraction. A 2-way ANOVA with Tukey's multiple comparison test was performed (p values * =≤ 0.05, ** =≤ 0.01, *** =≤ 0.001 and **** =≤ 0.0001) (*) indicate significance, when comparing HC with CF. (•) indicate significance between treatments within the same cell line.

DOI: https://doi.org/10.7554/eLife.49248.011

The following figure supplements are available for figure 5:

**Figure supplement 1.** Inhibition of amiloride-sensitive sodium channels modulates inflammation in cells with CF-associated mutations.
DOI: https://doi.org/10.7554/eLife.49248.012

**Figure supplement 2.** Inhibition of amiloride-sensitive sodium channels modulates inflammation in cells with CF-associated mutations.
DOI: https://doi.org/10.7554/eLife.49248.013

18 secretion at baseline and after LPS and ATP stimulation (p=0.0003) (*Figure 6B*). These data support the hypothesis that excessive ENaC-mediated Na$^+$ influx intrinsically drives NLRP3 inflammasome activation in CF.

## Discussion

Here we have described our findings that IL-1-type cytokines were elevated in both monocytes and serum of patients with CF, but two of the major inflammasome-independent cytokines, TNF and IL-6, were not significantly elevated in these patients. In contrast to patients with SAID, these data suggest that the proinflammatory cytokine response in CF has a predominant NLRP3 inflammasome constitution. Furthermore, on NLRP3 inflammasome activation, IL-18 secretion was upregulated in the CF-associated mutant cell lines, IB3-1 (p<0.0001) and CuFi-1 (p<0.0001) relative to the BEAS-2B control, and these levels were reduced by treatment with small molecule inhibition of NLRP3 inflammasome signalling, thereby confirming the NLRP3 inflammasome as a major source of the elevated IL-18 inflammatory cytokine in these cells. Perhaps the most notable feature is the uncovering of a molecular link between enhanced ENaC-dependent Na+ influx, observed in cells with CF-associated mutations, and the exacerbated NLRP3 inflammasome activation. By pretreating these cells with CF-associated mutations with small molecule inhibitors of ENaC, the exaggerated IL-1-type cytokine response in vitro was diminished (*Figure 7*).

Bronchial epithelial cell (BEC) lines can produce significant quantities of IL-18 on stimulation but secrete only negligible amounts of basal or stimulated IL-1β compared to hematopoietic cells (*Tang et al., 2012*; *Peeters et al., 2013*; *Gillette et al., 2013*). In human BEC models, rhinovirus has been associated with IL-1β release, a process accentuated by dual priming with both ATP and polyinosine polycytidylic acid (Poly I:C, which interacts with TLR3) (*Piper et al., 2013*; *Shi et al., 2012*). In general, IL-18 is the predominant cytokine secreted by non-myeloid cells, which are also capable of having activated inflammasomes (*Okazawa et al., 2004*). This contrasts with PBMCs, which produce large amounts of IL-1β, from both patients with CF and healthy controls (*Tang et al., 2012*). The differential secretion of IL-18 and IL-1β, observed in this study is noteworthy, with IL-1β secretion being undetectable in HBEC lines, under the conditions used. This disparity may reflect the function of these two inflammasome-processed zymogens; IL-18 is a cytokine that induces recruitment of neutrophils and Th17 differentiation, as well as IFNγ secretion, whereas IL-1β is an intrinsically more destructive cytokine, acting systemically to induce fever, proliferation, differentiation, apoptosis and sensitivity to pain. IL-1β secretion is tightly regulated by a highly controlled process, involving its own gene expression as well as inflammasome priming and assembly, whereas IL-18 is constitutively

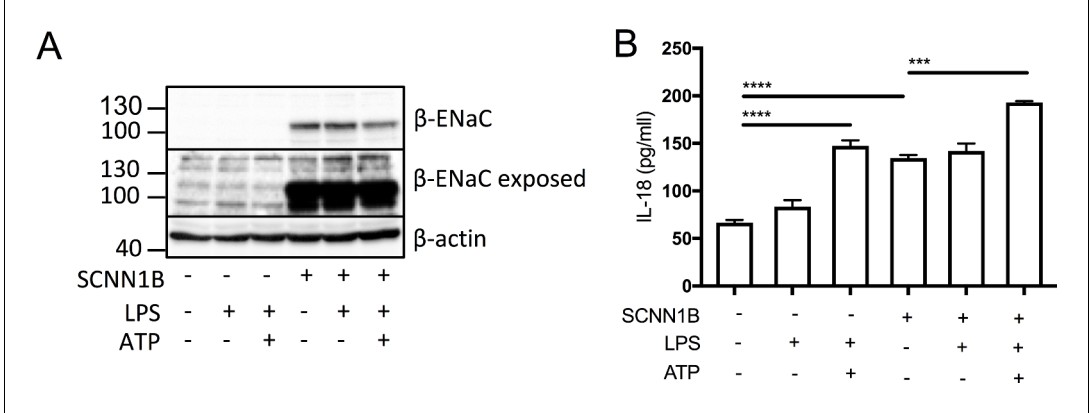

**Figure 6.** SCNN1B over-expression in BEAS-2B cells increases pro-inflammatory cytokine secretion. BEAS-2B cells were transiently transfected with 10μg SCNN1B cDNA (+) or a pcDNA3.1 vector only control (-) for 48 hr then stimulated with LPS (10 ng/mL, 4 hr) and ATP (5 mM) for the final 30 min (n = 3 independent experiments). Cells were lysed and immunoblotted for β-ENaC and β-actin a). ELISA assays were used to detect IL-18 in the supernatant b) fraction. A 2-way ANOVA with Tukey's multiple comparison test was performed (p values * =≤ 0.05, ** =≤ 0.01, *** =≤ 0.001 and **** =≤ 0.0001).
DOI: https://doi.org/10.7554/eLife.49248.014

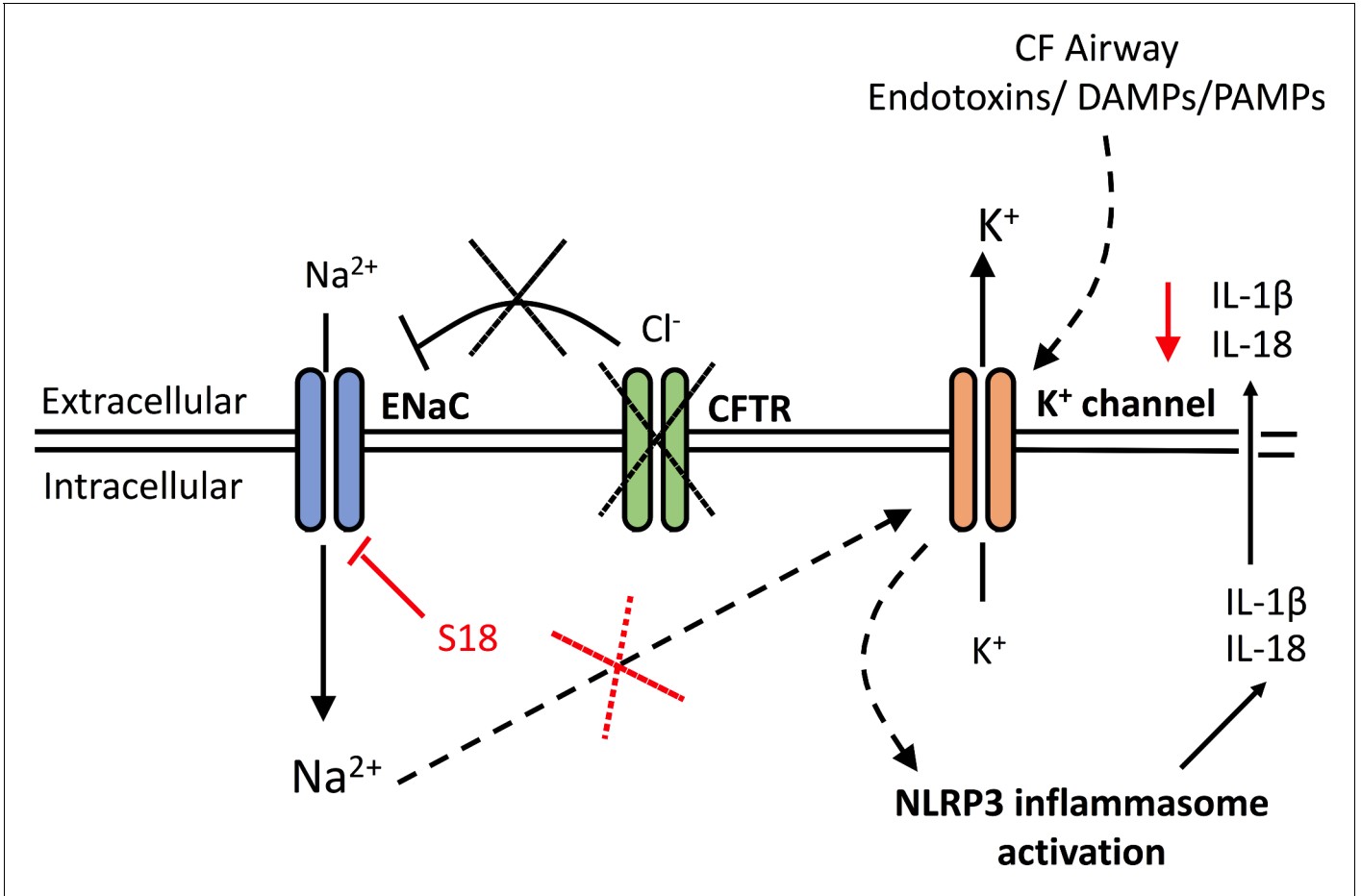

**Figure 7.** A schematic diagram of the proposed excessive NLRP3 inflammasome activation observed in individuals and cells with CF-associated mutations. Without functional CFTR, inhibition of ENaC currents is diminished leading to increased intracellular Na+ levels. Dysregulation of ENaC-dependent $Na^{2+}$ influx leads to increased $K^+$ efflux (via unknown mechanism) and NLRP3 inflammasome activation, with subsequent release of IL-1β and IL-18. In the CF airway, $K^+$ efflux is exacerbated upon K+ channel stimulation by endotoxins, DAMPs or PAMPs, leading to aberrant NLRP3 inflammasome activation and excessive IL-1β and IL-18 secretion. Blocking ENaC currents with S18 peptide restores Na+ and $K^+$ levels which reduces NLRP3-mediated production of IL-1β and IL-18.

DOI: https://doi.org/10.7554/eLife.49248.015

expressed and only depends on the two signals for inflammasome priming and assembly. In addition, IL-1β is rapidly sequestered, or secretion, by IL-1Ra and soluble IL-1RI and –RII, once secreted, which makes IL-1β particularly difficult to assay and detect. The biological property of IL-1β underlies the severe phenotype of the rare autoinflammatory disease, deficiency of the interleukin-1–receptor antagonist (DIRA), caused by homozygous mutations in the IL1RN gene, which encoding the IL-1Ra protein (*Aksentijevich et al., 2009*).

The primary consequence of mutations in the CFTR gene is defective CFTR anion transport and upregulation of $Na^+$ transport through dysregulation of ENaC (*Muraglia et al., 2019*; *Peckham et al., 1997*). Under in vivo conditions, patients with CF have a more negative baseline airway and nasal potential difference compared to HC and shows an enhanced depolarising response to amiloride, reflecting the inhibition of excessive ENaC mediated $Na^+$ influx (*Solomon et al., 2018*). In airway epithelial cells, increased basolateral Na/K-ATPase activity has also been reported (*Peckham et al., 1997*). We hypothesised that dysregulation of $Na^+$ transport might influence NLRP3 inflammasome activation by increasing $K^+$ efflux, a principal trigger for NLRP3 activation (*Schorn et al., 2011*; *Muñoz-Planillo et al., 2013*; *Katsnelson and George, 2013*; *Katsnelson et al., 2015*; *Di et al., 2018*; *Qu et al., 2017*; *Zhu et al., 2017*; *Rivers-Auty and Brough, 2015*). We confirmed a dysregulation of $Na^+$ and $K^+$ transport by examining primary innate

immune cells and HBEC lines with CF-associated mutations in vitro and discovered that excessive ENaC-mediated Na$^+$ flux, measured indirectly by dependent changes of fluorescence signal, correlate with an increased K$^+$ efflux, an activating signal of the NLRP3 inflammasome and downstream IL-18 and IL-1β secretion. We also show that the systemic serum cytokine signature from patients with CF is comparable to patients diagnosed with SAID, characterised by release of proinflammatory IL-1β and IL-18, which are associated with inflammasome activation. On exploration of the inflammation involved, we found a LPS-induced hyper-responsive NLRP3 inflammasome, exclusively in cells with CF-associated mutations, comparable to monocytes from patients with SAID. It should be noted that the SAID patient cohort is comprised of a variety of autoinflammatory diseases that have differing molecular pathophysiology and varying degrees of innate driven inflammation. This NLRP3 inflammasome activation extended beyond IL-18 and IL-1β secretion, with increased propensity for pyroptotic cell death and associated release of NLRP3 inflammasome components, such as ASC, and induction of IFN-γ secretion in the PBMC population. This type of inflammation has similarities to autoinflammation, particularly the IL-1/IL-18 inflammasome signature observed in serum and also present in vitro. The significantly elevated levels of ASC specks in CF sera (*Figure 3D*), in addition to the proinflammatory IL-1-type cytokine signature (*Figure 3A–C*), suggests the presence of a NLRP3 inflammasome agonist. A wide range of possible candidates for such an agonist(s) include infectious pathogens (either CF or non-CF related), the patients' metabolic milieu or, indeed, other unknown agonists. Further work will be necessary to decipher the complex molecular mechanisms involved. This study does not suggest that inflammation is independent of infections in individuals with CF but that the response to said infections is intrinsically dysregulated and predisposed to excessive and inappropriate degrees of inflammation.

Cystic fibrosis is a monogenic disease, yet consists of varying degrees of pathology depending on the precise CFTR mutation and the extent to which the encoded CFTR protein is subsequently transcribed, translated, folded within the ER, expressed on the plasma membrane, its stability on the surface and its ability to conduct chloride and bicarbonate. There are well documented examples of different classes of CFTR mutations that fail at each one of the above stages of CFTR expression and function. The IB3-1 (ΔF508/W1282X) cell line is associated with the greatest amounts of inflammation. The W1282X mutation is associated with severe disease clinically, that can be attributed to little to no protein expression. Inflammation in the IB3-1 cell line in vitro does not correlate with ENaC protein expression. The underlying mechanism of an ENaC/NLRP3 axis driving inflammation in cells with CF-associated mutations may not be common to all mutation classes. Notably, all of the cells (monocytes and epithelia) had at least one ΔF508 allele. Whether the loss of control of ENaC expression and function observed in ΔF508 CFTR expressing cells is unique to this more common mutation will require further study. It is important to note that inhibition of ENaC alleviated NLRP3 inflammasome-mediated inflammation in all cell lines (*Figure 5E–F*). While inflammation in CF is believed to be driven predominantly by infection, many CFTR mutant animal models have shown that airway inflammation and bronchiectasis can occur under sterile conditions (*Montgomery et al., 2017*; *Keiser et al., 2015*; *Rao and Grigg, 2006*). For instance, in a CFTR knockout ferret model, inflammation, bronchiectasis and mucus accumulation can develop in the absence of infection (*Rosenow, 2018*). A recent study from the Australian Respiratory Early Surveillance Team for CF highlights the association between pulmonary inflammation and structural lung disease in young children with and without infection (*Rosenow et al., 2019*). IL-1β is detectable in bronchoalveolar lavage (BAL) fluid from children with CF and is strongly correlated with neutrophil counts, independent from detectable infection (*Montgomery et al., 2018*). The increased IL-1β neutrophil production in BAL fluid from patients with CF appears to be driven via NLRP3 (*McElvaney et al., 2018*).

Autoinflammation is defined as an exaggerated inflammatory response, driven by dysregulated innate immune cells, in the absene of antigen-driven T-cells, B-cells, or associated autoantibodies (*Peckham et al., 2017*; *McDermott et al., 1999*; *McDermott and Aksentijevich, 2002*; *McGonagle and McDermott, 2006*; *Stoffels and Kastner, 2016*). Adaptive immune cells may be recruited in response to the downstream consequences of autoinflammation, with increased susceptibility to infection, and progression to autoimmunity and hyperinflammation (*Wekell et al., 2016*). In fact, we have shown that endoplasmic reticulum (ER) stress present in CF innate immune cells, including neutrophils, monocytes and M1 macrophages, causes an exaggerated inflammatory response (*Lara-Reyna et al., 2019*). Based on these data and cited literature, CF displays features of autoinflammatory disease, in part driven by aberrant ionic fluxes and recurrent infections. The

NLRP3 inflammasome can be primed by proinflammatory cytokines, such as TNF, although bacterial components do provide a far more potent stimulus (*Swanson et al., 2019*; *McElvaney et al., 2019*).

Targeting the exaggerated inflammatory response without simultaneously predisposing individuals to infection remains an elusive goal which carries inherent (potential) risk. This was highlighted by the termination of a study investigating a leukotriene B$_{4-}$ (LTB$_4$) receptor antagonist for treatment of lung disease in CF following a disproportionate incidence of respiratory serious adverse events (*Konstan et al., 2014*). Treating inflammation remains a priority and multiple trials targeting various anti-inflammatory pathways are ongoing (*Cystic Fibrosis Foundation, 2017*). There is evidence in the literature of endotoxin tolerant monocytes from patients with CF, with reduced cytokine expression in response to repetitive endotoxin exposure (*del Campo et al., 2011*; *del Fresno et al., 2009*; *del Fresno et al., 2008*). However, we propose that peripheral monocytes such as those used in this study, are not constantly exposed to common pathogenic antigens that exist within the lung microenvironment during infections. In fact, one may propose a scenario of trained immunity, where innate immune cells hyper-respond to the recurrent infections with greater destructive consequences. Endotoxins are not the only inflammatory stimulus for NLRP3 inflammasome priming that exist in the inflammatory milieu of the CF lung. Epithelial derived cytokines, neutrophilic extracellular traps (NETs), necrotic cell death and subsequent damage associated molecular patterns (DAMPs) may all induce transcription of IL-1β/IL-18 and other inflammasome components, independent of endotoxins. A caveat of our study is that we have not tested the full array of DAMPs and pathogen associated molecular patterns (PAMPs) that exist in the CF lung that may trigger the intrinsic predisposition to NLRP3 inflammasome activation observed in this study.

The data here suggest that IL-18 may be a useful therapeutic target, by preventing adaptive cell airway infiltration whilst maintaining a potent IL-1β response to infection (*Gabay et al., 2018*); however, NLRP3 activation exercises a protective role in animal models of induced colitis (*Allen et al., 2010*), running contrary to the expectation that reduced NLRP3 expression might reduce inflammation in the bowel. Furthermore, IL-18 has an epithelial protective role in promoting repair of gut epithelium (*Zaki et al., 2010*), and more ex vivo studies and CF disease models are required to elucidate the benefits or hazards of IL-18 blockade. Anakinra is a viable therapeutic option for CF, particularly with its short half-life and daily treatment regime allowing a more controlled dosage during any inevitable infections. In fact a phase IIa clinical trial to evaluate safety and efficacy of subcutaneous administration of anakinra in patients with cystic fibrosis is in progress (EudraCT Number: 2016-004786-80). It is notable that inhibition of TLR4 signalling was the most effective means of blocking NLRP3 activation in our study, which suggests that targeting this pathway may be a therapeutic option in CF (*Keeler et al., 2019*; *Greene et al., 2008*). These novel data, along with our observation that decreased intracellular K$^+$ levels upon stimulation with ATP in cells with CF-associated mutations, correlates with the characteristic and excessive ENaC-mediated Na$^+$ transport, suggest that CF-associated mutations lower the threshold for NLRP3 assembly, rather than priming this key intracellular component of innate immune defenses.

Through inhibition of amiloride-sensitive Na$^+$ channels, we were able to reduce NLRP3 inflammasome activation, and associated IL-18 and IL-1β secretion, in vitro. Notably, ENaC inhibition did not modulate IL-18 and IL-1β secretion in monocytes from individuals with SAID, indicating the proposed ENaC-NLRP3 axis is unique to CF-associated mutations. These findings highlight the importance of excessive ENaC-mediated intracellular Na$^+$ as a disease mechanism in CF, and also highlight its potential as a therapeutic target. Targeting amiloride-sensitive Na$^+$ channels such as ENaC to restoring airway surface liquid and mucociliary clearance, has been previously attempted, using amiloride in the 1990 s, with little efficacy, due to amiloride's short half-life and limited effectiveness (*Graham et al., 1993*).

In conclusion, we have shown that hypersensitive NLRP3 inflammasome activation in CF induces proinflammatory serum and cellular profiles. Overexpression of β-ENaC, in the absence of CFTR dysfunction as well as dysregulation of amiloride-sensitive Na$^+$ channel activity in CF, further potentiates LPS-induced NLRP3 inflammasome activity. Both ENaC and NLRP3 are potential therapeutic targets for reducing inflammation in patients with CF.

# Materials and methods

## Key resources table

| Reagent type (species) or resource | Designation | Source or reference | Identifiers | Additional information |
|---|---|---|---|---|
| Antibody | Rabbit polyclonal anti-SCNN1B | Avia Systems Biology, San Diego | Cat# ARP72375_P050; RRID: AB_2811256 | WB 1:500 |
| Antibody | Goat polyclonal anti-Rabbit IgG (H+L) Poly-HRP Secondary Antibody | ThermoFisher Scientific | Cat# 32260; RRID: AB_1965959 | WB 1:4000 |
| Antibody | Rabbit polyclonal anti-actin-β | GeneTex | Cat# GTX109639, RRID: AB_1949572 | WB 1:20:000 |
| Antibody | Mouse monoclonal Phycoerythrin anti-ASC (TMS-1) | Biolegend | Cat# 653903, RRID: AB_2564507 | 5 µL/ ml |
| Cell line (*Homo-sapiens*) | BEAS-2B cell line | ATCC | ATCC CRL-9609 | |
| Cell line (*Homo-sapiens*) | IB3-1 | ATCC | ATCC CRL-2777 | |
| Cell line (*Homo-sapiens*) | CuFi-1 cell line | ATCC | ATCC CRL-4013 | |
| Cell line (*Homo-sapiens*) | CuFi-4 cell line | ATCC | ATCC CRL-4015 | |
| Commercial assay or kit | MycoAlertTM | Lonza | Cat# LT07-118 | |
| Biological samples (*Homo-sapiens*) | Human Blood Samples | St James's University Hospital | Health Research Authority REC reference 17/YH/0084 | |
| Chemical compound, drug | Lymphoprep | Axis Shield | Cat# 1114544 | |
| Chemical compound, drug | Pan Monocyte Isolation Kit, human | Miltenyi Biotec | Cat# 130-096-537 | |
| Chemical compound, drug | Lipopolysacchride Ultrapure EK | InvivoGen | Cat# tlrl-eklps | 10ng/ml |
| Chemical compound, drug | MCC950 | Cayman Chemical | Cat# CAY17510-1 | 15 nM, 1 hr |
| Chemical compound, drug | YVAD | InvivoGen | Cat# inh-yvad | 2 µg/ mL, 1 hr |
| Chemical compound, drug | OxPAPC | InvivoGen | Cat# tlrl-oxp1 | 30 µg/ mL, 1 hr |
| Chemical compound, drug | Amiloride (hydrochloride) | Cayman Chemical | Cat# 26295 | 10 µM, 100 µM, 1 hr |
| Chemical compound, drug | SPLUNC1-derived peptide, S18 | Gift from Spyryx Biosciences, Inc | | 25 µM, 4 hr |
| Chemical compound, drug | 5-(N-ethyl-N-isopropyl)-Amiloride (EIPA) | Cayman Chemical | Cat# 1154-25-2 | 10 µM, 1 hr |
| Chemical compound, drug | Ouabain | Torcis Bioscience | Cat# 630-60-4 | 100 nM, 24 hr |
| Chemical compound, drug | ATP | InvivoGen | Cat# tlrl-atpl | 5 mM, 30 min |
| Chemical compound, drug | poly(dA:dT) dsDNA | InvivoGen | Cat# tlrl-patn | 1 µg/ mL, 1 hr |
| Chemical compound, drug | TcdB | Cayman Chemical | Cat# CAY19665-50 | 10 ng/mL, 1 hr |

*Continued on next page*

*Continued*

| Reagent type (species) or resource | Designation | Source or reference | Identifiers | Additional information |
|---|---|---|---|---|
| Chemical compound, drug | Flagellin | InvivoGen | Cat# tlrl-pbsfla | 10 ng/mL, 1 hr |
| Commercial assay or kit | Pierce BCA Protein Assay Kit | ThermoFisher Scientific | Cat# 23225 | |
| Chemical compound, drug | PhosSTOP | Merck | Cat# 4906845001 | |
| Chemical compound, drug | Pierce Protease Inhibitor Mini Tablets | ThermoFisher Scientific | Cat# A32955 | |
| Chemical compound, drug | Immobilon Western Chemiluminescent HRP Substrate | Merck | Cat# WBKLS0500 | |
| Commercial assay or kit | IL-1 beta Human Matched Antibody Pair | ThermoFisher Scientific | Cat# CHC1213 | Assay sensitivity < 31.2 pg/mL |
| Commercial assay or kit | IL-18 Human Matched Antibody Pair | ThermoFisher Scientific | Cat# BMS267/2MST | Assay sensitivity 78 pg/mL |
| Commercial assay or kit | IL-6 Human Matched Antibody Pair | ThermoFisher Scientific | Cat# CHC1263 | Assay sensitivity 15.6 pg/mL |
| Commercial assay or kit | TNF alpha Human Matched Antibody Pair | ThermoFisher Scientific | Cat# CHC1753 | Assay sensitivity < 15.6 pg/mL |
| Commercial assay or kit | IL1RA Human Matched Antibody Pair | ThermoFisher Scientific | Cat# CHC1183 | Assay sensitivity < 31.2 pg/mL |
| Chemical compound, drug | (TMB) substrate solution | Sigma | Cat# T0440 | |
| Commercial assay or kit | Caspase-1 Colorimetrix Assay | R and D Systems | Cat# BF15100 | |
| Commercial assay or kit | High-Capacity cDNA Reverse Transcription Kit | ThermoFisher Scientific | Cat# 4368814 | |
| Recombinant DNA reagent | SCNN1B cDNA plasmid | Addgene | Cat# 83429 | |
| Recombinant DNA reagent | pcDNA3.1 cDNA plasmid | Gift from N.M Hooper, Manchester | | |
| Chemical compound, drug | sodium-sensitive molecule SBFI | ThermoFisher Scientific | Cat# S-1263 | 10 mM, 100 min |
| Chemical compound, drug | potassium-sensitive molecule PBFI | ThermoFisher Scientific | Cat# P-1266 | 10 mM, 100 min |
| Chemical compound, drug | Pluronic F-127 | Sigma | Cat# P2443 | |
| Software, algorithm | GraphPad Prism7 | Graphpad software | | |

## Clinical characteristics of patients

Patients with CF, systemic autoinflammatory diseases (SAID), non-CF bronchiectasis (NCFB) and healthy controls (HC) were recruited from the Department of Respiratory Medicine and Research laboratories at the Wellcome Trust Benner Building at St James's Hospital. All SAID patients were on Anakinra treatment, when blood samples were obtained. The study was approved by Yorkshire and The Humber Research Ethics Committee (17/YH/0084). Informed written consent was obtained from all participants at the time of the sample collection. Demographics are shown in *Supplementary file 1*. All CF donors were F508del/F508del homozygous (n = 30) with no sign of infection. Three of the patients with NCFB had primary ciliary dyskinesia (PCD) and one patient with NCFB had an unknown genotype. All patients with a SAID had characterised mutations in a known disease-causing gene (Tumor Necrosis Factor Receptor Associated Periodic Syndrome (TRAPS) n = 2, Muckle-Wells n = 2,

A20 haploinsufficiency n = 1, Pyrin-Associated Autoinflammation with Neutrophilic Dermatosis (PAAND) n = 1, Familial Mediterranean Fever (FMF) n = 2, Hyper IgD Syndrome (HIDS) n = 2 and Schnitzler syndrome n = 1).

## PBMC and monocyte isolation

Peripheral blood mononuclear cells (PBMCs) were isolated from whole blood using the density gradient centrifugation. Whole blood were mixed with equal volume of PBS, carefully layered onto of Lymphoprep (Axis-Shield, Dundee, UK) and centrifuged at 1100*xg* for 20 min without brakes. The white buffy layer was removed and washed twice in PBS by centrifuging at 1100*xg* for 10 min. PBMC pellet was resuspended in complete RPMI medium (RPMI medium containing 10% heat inactivated foetal bovine serum, 50 U/ml penicillin, 50 µg/ml streptomycin).

Monocytes were isolated by negative selection from PBMCs using the monocyte isolation kit II (Miltenyi Biotec GmbH, Bergisch Gladbach, Germany). Pelleted PBMCs were resuspended in 30 µl of buffer per $10^7$ cells (autoMACS Rinsing Solution containing 0.5% BSA). This was mixed with 10 µl FcR Blocking Reagent followed by 10 µl of biotin-conjugated antibodies and incubated at 4°C for 10 min. Next 30 µl of buffer were added together with 20 µl of anti-biotin microbeads and incubated for an additional 15 min at 4°C. This whole mixture was washed with 2 ml of buffer and centrifuged at 300 *xg* for 10 min. The cell pellet was resuspended in 500 µl of buffer and past down a MS column (Miltenyi Biotec GmbH, Bergisch Gladbach, Germany) on a magnetic stand. PBMCs ($2 \times 10^6$/ ml) and monocytes ($1 \times 10^6$/ ml) were allowed to adhere overnight prior to experimentation.

## Cell line culture

Human cell lines BEAS-2B (ATCC CRL-9609), CuFi-1 (ATCC CRL-4013), CuFi-4 (ATCC CRL-4015) and IB3-1 (ATCC CRL-2777) were purchase from ATCC (UK) which ensures STR profiling of the cell lines used. BEAS-2B and IB3-1 were cultured in LHC basal medium (Thermo Fisher Scientific, Loughborough, UK) supplemented with 10% FBS, 50 U/ml penicillin and 50 µg/ml streptomycin). CuFi-1 and CuFi-4 were grown on Cell+ surface plates or flasks (Sarstedt, Leicester, UK) with LHC-9 medium (Thermo Fisher Scientific, Loughborough, UK). All cells were cultured in a humidified incubator at 37°C, 5% $CO_2$. Cells were used at $1 \times 10^6$/ ml. Cell lines were routinely tested for mycoplasma using MycoAlertTM Mycoplasma Detection Kit Lonza catalog#: LT07-118, and were all negative.

## Cell stimulations

Cells were pre-treated with the following compounds where indicated prior to NLRP3 stimulation; MCC950 (15 nM, Cayman Chemical, Cambridge, UK) for 1 hr, YVAD (2 µg/ mL, Invivogen, San Diego, California) for 1 hr, OxPAPC (30 µg/ mL, Invivogen) for 1 hr, Amiloride (10 µM, 100 µM, Cayman Chemical, Cambridge, UK) for 1 hr, EIPA (10 µM, Cayman Chemical) for 1 hr, SPLUNC1-derived peptide, S18 (25 µM, gift from Spyryx Biosciences, Inc) for 4 hr or ouabain (100 nM, Cayman Chemical, Cambridge, UK) for 24 hr. Inflammasome stimulation was achieved using either LPS (10 ng/mL, Ultrapure EK, Invivogen) for 4 hr with the addition of ATP (5 mM, Invivogen, San Diego, California) for the final 30 min of stimulation, poly(dA:dT) dsDNA (1 µg/mL with Lipofectamine 2000, Invivogen, San Diego, California) for 1 hr, TcdB (10 ng/mL, Cayman Chemical, Cambridge, UK) for 1 hr or flagellin (10 ng/mL with Lipofectamine 2000, Invivogen, San Diego, California) for 1 hr for the final 1 hr of the LPS stimulation. ATP was dissolved in pre-warmed at 37°C medium (100 mM stock) and immediately (~2 min) added to the cells. All incubations were done in a humidified incubator at 37°C, 5% $CO_2$. Supernatant, RNA and protein were collected and stored immediately following stimulation.

## Cytokine quantification using ELISA

Cytokines from patient sera and cell cultured media were detected by ELISAs (IL-1 beta Human Matched Antibody Pair, human IL-18 Matched Antibody Pair, IL1RA Human Matched Antibody Pair, TNF alpha Human Matched Antibody Pair and IL-6 Human Matched Antibody Pair) (ThermoFisher Scientific, Loughborough, UK), as per the manufactures recommendations. In general, ELISA plates were coated with 100 µl cytokine capture antibody in PBS overnight at 4°C. The plates were washed three times with PBST (PBS containing 0.5% Tween 20) and the wells blocked in 300 µl assay buffer (0.5% BSA, 0.1% Tween 20 in PBS) by incubating for 1 hr. The plates were washed twice with PBST and 100 µl of sera/culture supernatants, together with appropriate standards, were added to wells

in duplicates. Immediately 50 µl of detection antibody were added to all wells and incubated for 2 hr. After the incubation the plates were washed five times with PBST and 100 µl of tetramethybenzidine (TMB) substrate solution (Sigma, Poole, UK) were added to all wells and incubated for 30 min. Colour development was stopped by adding 100 µl of 1.8N $H_2SO_4$. And absorbance measured at 450 nm and reference at 620 nm. Note all incubation steps were done at room temperature with continual shaking at 700 rpm. All data points are an average of duplicate technical replicates for each independent experiment.

## ASC protein aggregates (specks)

Methodology as previously published (*Rowczenio et al., 2018*). Briefly, patients' sera or culture media were incubated with 5 µL of phycoerythrin anti-ASC (TMS-1) antibody (Biolegend, London, UK) for 1 hr, and analysed on the LSRII flow cytometer instrument (BD Biosciences, California).

## Caspase-1 activity

A colorimetric assay (Caspase-1 Colorimetrix Assay, R and D Systems, Abingdon, UK) measured caspase-1 activity, via cleavage of a caspase-specific peptide conjugated to a colour reporter molecule, p-nitroalinine (pNA), performed on protein lysates and serum. All data points are an average of duplicate technical replicates for each independent experiment. Protein concentrations in the lysate were determined by BCA assay.

## Detection of mRNA by RT-qPCR

Cells were washed in PBS, pelleted and immediately lysed in 1 ml TRIzol Reagent (Ambion Life technologies, Paisley, UK) and RNA extracted using the PureLink RNA mini kit (Ambion, Life technologies, Paisley, UK). Chloroform (200 µl) were added to each sample and mixed vigorously for 15 s and left to stand for 2 min at room temperature. These were centrifuged at 12000 *xg* for 15 min at 4°C. The top clear phase was transferred to a fresh tube and mixed with equal volume of 70% ethanol. This mixture was transferred to a spin cartridge, with a collection tube, and centrifuged at 12000 *xg* for 15 s at room temperature. The waste was disposed of and the spin cartridge was centrifuged one more time. The spin cartridge was washed 700 µl of wash buffer I and centrifuged at 12000 *xg* for 15 s. A second wash with 500 µl wash buffer II (containing ethanol) were added to the spin cartridge and centrifuged at 12000 *xg* for 15 s followed by further spin for 1 min. RNA was recovered by adding 30 µl of RNase free water to the spin cartridge, incubated for 1 min and centrifuged for 2 min.

The High Capacity cDNA Reverse Transcription kit (Applied Biosystems, California) was used to convert the RNA to cDNA according to the manufacturer's instructions. TaqMan assays were done in the QuantStudio 5 Real-Time PCR instrument (Thermo Fisher Scientific, Loughborough, UK).

The Taqman primers used in this study are detailed below:

| Gene | Assay ID | Dye |
| --- | --- | --- |
| SCNN1B | Hs01548617_m1 | FAM |
| IFNG | Hs00989291_m1 | FAM |
| HPRT1 | Hs02800695_m1 | FAM |

## Transient transfection

BEAS-2B cells were transiently transfected with 10 µg of sodium channel epithelial one beta subunit (SCNN1B) cDNA (Addgene, Teddington, UK) or pcDNA3.1 vector only control (gift from Professor NM Hooper, Manchester University) using Lipofectamine 2000 (Thermo Fisher Scientific, Loughborough, UK) for 48 hr, as per manufacturers' instructions. Cells were harvested, lysed in RIPA buffer and protein concentrations were determined using the Pierce bicinchoninic acid (BCA) assay (Thermo Fisher Scientific, Loughborough, UK).

## Western blotting

Samples were made up in dissociation buffer [1x dissociation buffer (100 mM Tris-HCl, 2% (w/v) sodium dodecyl sulfate, 10% (v/v) glycerol, 100 mM dithiothreitol, 0.02% (w/v) bromophenol blue, pH 6.8] and heated at 95°C for 5 min, and equal protein concentration was loaded and resolved by 10% SDS-PAGE on Tris-glycine gels and then transferred to Hybond PVDF membranes (GE Healthcare, Buckinghamshire, UK). Following electrotransfer in Towbin buffer (25 mM Tris, 192 mM glycine, and pH 8.3, 20% methanol) at 100 V for 1 hr, the membranes were blocked for 1 hr in blocking solution (PBS containing 0.1% Tween 20% and 5% (w/v) non-fat milk). After three washes in PBST (PBS with 0.5% Tween 20), primary antibodies were incubated with PVDF membrane overnight at 4°C. The membrane was washed three times with PBST and secondary antibody-HRP conjugate were added and incubated for 2 hr with constant rocking at room temperature. The membrane was washed five times with PBST and 3 ml of ECL detection system (Immobilon chemiluminescent HRP substrate, Millipore, UK) was added onto the membrane for 5 min, before being imaged with the ChemiDoc Imaging system (Bio-Rad, Hertfordshire, UK). Primary antibodies used: rabbit anti-SCNN1B (Avia Systems Biology, San Diego; 1/500 dilution), rabbit anti-actin-β (GeneTex, Nottingham, UK) at 1/20000 dilution. Secondary antibodies used: anti-rabbit IgG horseradish peroxidase-conjugated (Cell Signalling Technology, Hertfordshire, UK) were diluted at 1/4000. All antibodies were diluted in PBS containing 0.1% Tween 20% and 2% BSA.

## Fluorometric determination of na+ and K+ concentration

Na$^+$ and K$^+$ sensitive dyes, SBFI (S-1263) and PBFI (P-1266) (Molecular Probes, Paisley, UK), respectively, were used as cell permeant selective ion indicators for the fluorometric determination of Na+ and K+ concentrations. Monocytes and HBECs were allowed to adhere in black 96-well cell culture plates overnight. Cells were then incubated with various stimulants, as indicated in figure legends, before being washed and incubated in the appropriate low serum media (see above) (1%) for 1 hr. The dyes (10 mM final concentration) were loaded with Pluronic F-127 (Sigma) and incubated for 100 min. All wells were washed with NaCl solution. ATP (5 mM) was then added to the wells prior to measuring fluorescence. Excitation at 344 nm and 400 nm with emission at 500 nm was measured immediately to calculate the percentage change in fluorescence compared to an untreated control. All data points are an average of duplicate technical replicates for each independent experiment.

## Statistics

All analyses were performed using GraphPad Prism v 7. Bar graphs were expressed as mean standard error of the mean (S.E.M). The Kruskal-Wallis test with Dunn's multiple comparison or the Mann Whitney test was performed when comparing non-parametric populations. A two-way ANOVA statistical test with Tukey's multiple comparison post-hoc analysis was performed when calculating variance between samples (p values * =$\leq$ 0.05, ** =$\leq$ 0.01, *** =$\leq$ 0.001 and ****=$\leq$0.0001). A p<0.05 was considered significant. Statistical tests used are indicated in the figure legends. p-values are measured using a 2-sided hypothesis.

## Acknowledgements

The authors would like to thank all the patients and research nurses, particularly Lindsey Gillgrass and Anne Wood, of the Adult Cystic Fibrosis Unit at St. James's Hospital, Leeds. This work is supported by a grant (SRC009) from the Cystic Fibrosis Trust, a 110 Anniversary University of Leeds Scholarship (TS), LIRMM scholarship and CONACyT (SLR).

## Additional information

### Funding

| Funder | Grant reference number | Author |
| --- | --- | --- |
| Cystic Fibrosis Trust | SRC009 | Heledd H Jarosz-Griffiths<br>Chi Wong<br>Jonathan Holbrook<br>Fabio Martinon<br>Sinisa Savic<br>Daniel Peckham<br>Michael F McDermott |
| University of Leeds | 110 University Scholarship | Thomas Scambler |
| Consejo Nacional de Ciencia y Tecnología | CONACyT | Samuel Lara-Reyna |

The funders had no role in study design, data collection and interpretation, or the decision to submit the work for publication.

### Author contributions

Thomas Scambler, Conceptualization, Data curation, Formal analysis, Investigation, Methodology, Writing—original draft, Writing—review and editing; Heledd H Jarosz-Griffiths, Data curation, Formal analysis, Investigation, Methodology, Writing—original draft, Writing—review and editing; Samuel Lara-Reyna, Shelly Pathak, Data curation, Formal analysis; Chi Wong, Data curation, Formal analysis, Project administration; Jonathan Holbrook, Writing—review and editing; Fabio Martinon, Conceptualization, Supervision, Investigation, Visualization, Writing—review and editing; Sinisa Savic, Conceptualization, Investigation, Visualization, Writing—original draft, Writing—review and editing; Daniel Peckham, Conceptualization, Resources, Supervision, Funding acquisition, Investigation, Visualization, Writing—original draft, Project administration, Writing—review and editing; Michael F McDermott, Conceptualization, Resources, Funding acquisition, Investigation, Writing—original draft, Project administration, Writing—review and editing

### Author ORCIDs

Thomas Scambler (iD) https://orcid.org/0000-0003-2468-0218
Heledd H Jarosz-Griffiths (iD) https://orcid.org/0000-0001-5154-4815
Samuel Lara-Reyna (iD) https://orcid.org/0000-0002-9986-5279
Chi Wong (iD) https://orcid.org/0000-0003-2108-1615
Fabio Martinon (iD) http://orcid.org/0000-0002-6969-822X
Sinisa Savic (iD) http://orcid.org/0000-0001-7910-0554
Daniel Peckham (iD) https://orcid.org/0000-0001-7723-1868
Michael F McDermott (iD) https://orcid.org/0000-0002-1015-0745

### Ethics

Human subjects: Patients with CF, systemic autoinflammatory diseases (SAID), non-CF bronchiectasis (NCFB) and healthy controls (HC) were recruited from the Department of Respiratory Medicine and Research laboratories at the Wellcome Trust Benner Building at St James's Hospital. The study was approved by Yorkshire and The Humber Research Ethics Committee (17/YH/0084). Informed written consent was obtained from all participants at the time of the sample collection.

### Decision letter and Author response

Decision letter https://doi.org/10.7554/eLife.49248.019
Author response https://doi.org/10.7554/eLife.49248.020

## Additional files

### Supplementary files

• Supplementary file 1. Donor demographics. Patients with Cystic Fibrosis (CF), systemic autoinflammatory diseases (SAID), non-CF bronchiectasis (NCFB) and healthy controls (HC). All CF donors were F508del/F508del homozygous (n = 30) with no sign of infection. Three of the patients with NCFB had primary ciliary dyskinesia (PCD) and one patient with NCFB had an unknown genotype. All patients with a SAID had characterised mutations in a known disease-causing gene (Tumor Necrosis Factor Receptor Associated Periodic Syndrome (TRAPS) n = 2, Muckle-Wells n = 2, A20 haploinsufficiency n = 1, Pyrin-Associated Autoinflammation with Neutrophilic Dermatosis (PAAND) n = 1, Familial Mediterranean Fever (FMF) n = 2, Hyper IgD Syndrome (HIDS) n = 2 and Schnitzler syndrome n = 1). BMI: Body Mass Index; FEV: Forced expiratory volume; CRP: C-reactive protein.
DOI: https://doi.org/10.7554/eLife.49248.016

• Transparent reporting form
DOI: https://doi.org/10.7554/eLife.49248.017

### Data availability

All data generated or analysed during this study are included in the manuscript and supporting files.

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
