## [Decision Letter]

Thank you for submitting your article "ENaC-mediated sodium influx exacerbates NLRP3-dependent inflammation in Cystic Fibrosis" for consideration by *eLife*. Your article has been reviewed by three peer reviewers, including Jos van der Meer as the Reviewing Editor and Reviewer #1, and the evaluation has been overseen by Satyajit Rath as the Senior Editor. The following individual involved in review of your submission has agreed to reveal their identity: Siroon Bekkering (Reviewer #2).

The reviewers have discussed the reviews with one another and the Reviewing Editor has drafted this decision to help you prepare a revised submission.

The reviewers agree this is an interesting paper exploring NLRP3 inflammasome activation in Cystic fibrosis. In general, the experiments are well carried out and convincing.

The authors compare the IL-1 and IL-18 (as well as caspase-1 and ASC specks) in CF production directly to patients with Autoinflammatory syndromes (as well as to patients with non CF bronchiectases and Healthy controls), and in principle this gives a good impression of the magnitude of autoinflammation in CF.

The authors convincingly show that the responses in CF are NLRP3 inflammasome-specific as they are blocked by an NLRP3-specific small molecule inhibitor, and because AIM2, NLRC4, or Pyrin inflammasome responses are not elevated. Mechanistically, the authors find that there are significant changes in levels of intracellular Na^+^ (increased) and K^+^ (decreased) in CF monocytes and CF mutant HBECs when NLRP3 is activated. This is associated with increased expression of epithelial sodium channels (ENaC) suggesting that CFTR mutations drive the dysregulation of ENaC, which via sodium influx leads to increased potassium efflux,which is a central trigger of NLRP3 activation. Critically, using specific inhibitors of ENaC attenuates the hyperactivation of NLRP3 observed in CF monocytes and CF mutant HBECs, but does not affect healthy control cells.

Although the paper is generally well presented, there is some lack of precision, which is reflected by the large number of comments.

Essential revisions:

1) Is it possible that ER stress driven by CFTR mutations contributes to NLRP3 activation? Has this been evaluated by the authors?

2) In all experiments, the LPS or ATP only controls are missing. This is of special importance in the monocyte stimulation experiments, since monocytes can activate the inflammasome independent of ATP as shown by Gaidt et al., 2016. If the controls were included in the experiment but not shown, that should be mentioned or shown in the supplementary.

3) Figure 1: What is OxPac? Is this supposed to be OxPAPC?

4) As said the comparison with autoinflammatory syndromes (SAID) is a major asset to the paper, but it should be realised that SAID is a mixed bag. Although these are all IL-1beta disorders, the pathophysiology differs, as well as the magnitude of the inflammatory response. The authors could be a bit more critical on this.

5) In the Introduction, the rationale of the comparison with SAID and NCFB should be explained.

6) In line with comment #2, it is clear that there are some outliers in the SAID group in Figure 3 A and B. Are these the same patients? What was their underlying disorder? This information should be included at least in the legend to Figure 3.

7) Patients with SAID are often found to have ex-vivo IL-1beta production in the absence of an inflammatory stimulus. The question is whether this is the case in CF. It is a pity that in Figure C and D unstimulated cultures are not depicted. Are these measurements available? The blank measurements in Figure 2B are a bit of a concern as the IL-1beta concentrations are in the range of low dose LPS stimulation (and hence contamination).

8) In some experiments, all monocytes (CF, SAID and NCFB) are studied. In others, only monocytes from CF. The comparison with SAID monocytes is especially interesting in the Na/K experiments and for example the IFNgamma experiments. Why are they missing there?

9) IL-1beta measurements in serum are most often below or close to the detection level of the commercial ELISA assays. The authors should inform us about the level of detection with their use of the assay. This also holds for the other ELISAs used.

10) In line with the previous comment: the serum cytokine concentrations measured are extremely high, even for healthy controls. Please check whether the assays were performed correctly, and calculations done right. Usually concentrations of circulating IL-6 are usually around 1 pg/ml (and not 50-200), IL-1beta below 1 pg/ml (here values ranging from 5 – 90pg/ml are reported). Are CRP values known for the participants?

11) In subsection “Proinflammatory cytokines and ASC specks are elevated in CF sera, and are comparable to patients diagnosed with systemic autoinflammatory disease (SAID)” it suddenly becomes clear that the SAID patients were on Anakinra treatment. This should have been mentioned in the patient section.

12) In connection with the previous comment: by blocking the IL-1 receptor, Anakinra will inhibit IL-1beta induced by IL-1(β and α). In general, this effect is rather limited if one looks at circulating IL-1beta. Of course, secondary cytokines like IL-6 and IL-8 will be lower. Thus, the sentence” All SAID patients were on active recombinant IL-1Ra (anakinraR) therapy, which will have reduced serum IL-1b levels.” should be rephrased.

13) Subsection “Proinflammatory cytokines and ASC specks are elevated in CF sera, and are comparable to patients diagnosed with systemic autoinflammatory disease (SAID)”: how do the authors know that they detect endogenous IL-1Ra and not Anakinra?

14) It is hard to believe that the genotypes of the SAID patients (with the exception of the Schnitzler patient) are not known (Suppl Table).

15) In the PBMC stimulation experiments (in the Materials and methods as well as in the figure legends) details are missing. How long were the cells stimulated, how many cells, and how much stimulus?

16) The ATP preparation is missing in the Materials and methods section, how was ATP prepared for these experiments? This is of great importance as described by Stoffels et al., 2015. Please add methods.

17) Although the discussion on the disparity between IL-1 and IL-18 production is interesting (Discussion section), the reasons why the cell lines do not show IL-1beta production should also be discussed.

18) In the same vein, there should be some discussion of the responses of the different mutation HBEC cell lines. Generally, there seems to be the most significant effects with the IB3-1 line but this does not correlate with the level of b-ENaC observed by Western blotting (Figure 4G).

19) The Discussion section again starts with an introduction. Instead it should start mentioning the main findings of the paper, and deal with the aims that were set at the end of the Introduction. So at least skip the first paragraph and rephrase the sentences that follow.

20) The authors are very prudent discussing the potential therapeutic consequences (Discussion section). Anakinra would be worth trying, as it does not meet with major infectious complications (at least much less so than Canakinumab). In fact, it has been given successfully to patients with Chronic granulomatous disease (who also suffer from excess IL-1beta production, but at the same time have a serious immunodeficiency).

21) A scheme summarizing the mechanism and sequence of events would help the readers.

22) As a suggestion: the authors might discuss why CF patients that are constantly exposed to *Pseudomonas* spp do not show endotoxin-tolerant monocytes.

---

## [Author Response]

We thank the reviewers for their positive comments and for their thorough evaluation of the manuscript.

Essential revisions:1) Is it possible that ER stress driven by CFTR mutations contributes to NLRP3 activation? Has this been evaluated by the authors?

It is possible that ER stress is a contributing factor to NLRP3 activation. In a recent publication by our group, we have shown that ER stress is present in CF innate immune cells, affecting human bronchial epithelial cells (HBECs), neutrophils, monocytes and M1 macrophages. We found high levels of IL-6 and TNF in M1 macrophages, with an associated activation of the IRE1α-XBP1 pathway, which could be reversed by inhibition of the RNase domain of IRE1a (Lara-Reyna et al., 2019).

To expand on this, we have shown that inhibition of the RNase domain of IRE1α with 4μ8c in LPS/ATP stimulated CF monocytes reduces TNF levels significantly in healthy control (HC) (p=0.04, n=3) and in CF (p<0.0001, n=3), consistent with our observations in M1 macrophages. Interestingly, IL-1β levels were also significantly reduced in IRE1α inhibited CF monocytes following stimulation with LPS/ATP (p<0.001, n=3) but although reduced, did not reach significance in HC (P=0.772, n=3). This data is consistent with a recent report in which inhibition of IRE1α RNase domain with MKC8866 reduces IL1β levels in PBMCs stimulated with LPS/ATP (Talty et al., 2019). It seems that as well as modulating the unfolded protein response and managing ER stress, IRE1a signalling also promotes the efficiency of inflammasome assembly, and that blocking IRE1α seems to have a direct effect on inflammasome activation (Talty et al., 2019).

Our previous work shows that IRE1α mRNA levels are increased in CF relative to HC, and IRE1α protein levels are increased in HBEC’S with CF-associated mutations relative to wild-type controls (Lara-Reyna et al., 2019). We can conclude from our new data that ER stress driven by CFTR mutations may, in part, contribute to NLRP3 activation seen in CF, but further work would be required to establish whether this effect is due to ER stress, as a result of CFTR-dependent IRE1α, upregulation or another factor(s).

We have added the following into the Discussion section:

“In fact, we have shown that endoplasmic reticulum (ER) stress present in CF innate immune cells, including neutrophils, monocytes and M1 macrophages, causes an exaggerated inflammatory response.”

2) In all experiments, the LPS or ATP only controls are missing. This is of special importance in the monocyte stimulation experiments, since monocytes can activate the inflammasome independent of ATP as shown by Gaidt et al., 2016. If the controls were included in the experiment but not shown, that should be mentioned or shown in the supplementary.

We have included LPS only controls for HC, CF, SAID and NCFB as a supplementary figure (Figure 2—figure supplement 2A,B). The manuscript by Gaidt et al., 2016 suggests that LPS alone can activate NLRP3 inflammasome through upstream activation of TLR4-TRIF-RIPK1-FADD-CASP8. They show an increase in IL-1β when monocytes were stimulated for 14 hours with 2 μg/ml LPS. They include a titration (9 different concentrations) with LPS from 2 μg/ml to 200 fg/ml and detected IL-1β cytokine production by ELISA. Unfortunately, they do not indicate the specific concentrations used – at the fourth titration of LPS alone, LPS does not induce IL-1β production in the absence of nigerecin, suggesting that low doses of LPS does not activate the non-classical inflammasome activation pathway.

As part of our experiments we did stimulate our monocytes with LPS (10ng/ml for 4h) alone (but did not include the data in the original manuscript) and observed no increase in IL-1β or IL-18 in HC, CF or NCFB patients under these conditions in the absence of ATP.

We have included the graphs below in Figure 2—figure supplement 2A,B. And included the following comment in the text (subsection “Increased NLRP3-dependent IL-1β/ IL-18 secretion in human monocytes with CF-associated mutations”):

“Under basal conditions primary monocytes, isolated from HC and CF, showed no significant difference in the secretion of IL-18 and IL-1β cytokines (Figure 2A, B) or when monocytes were stimulated with LPS alone across all patient groups (Figure 2—figure supplement 1A,B)”.

3) Figure 1: What is OxPac? Is this supposed to be OxPAPC?

We have amended this to OxPAPC throughout the manuscript. OxPAPC is a TLR2 and TLR4 inhibitor.

4) As said the comparison with autoinflammatory syndromes (SAID) is a major asset to the paper, but it should be realised that SAID is a mixed bag. Although these are all IL-1beta disorders, the pathophysiology differs, as well as the magnitude of the inflammatory response. The authors could be a bit more critical on this.

In fact, the variety of autoinflammatory disorders, described in this manuscript, was deliberately chosen to demonstrate the broad range of pathophysiology within this rare inflammatory disease spectrum, as the reviewers rightly state, thereby offering a complete and fair comparison between SAID and CF. We have also added a short description of the differences, in both the genetics and inflammatory response of autoinflammatory disease, into the manuscript and how CF might theoretically fit into this spectrum.

We have included the following into the Discussion section:

“It should be noted that the SAID patient cohort is comprised of a variety of autoinflammatory diseases that have differing molecular pathophysiology (Savic et al., in press) and varying degrees of innate driven inflammation.”

5) In the Introduction, the rationale of the comparison with SAID and NCFB should be explained.

We have revised the Introduction to outline this comparison:

“In order to fulfil these aims, monocytes and epithelial cells with characterised CF-associated mutations are directly compared to cohorts of NCFB and SAID. The NCFB cohort comprises of individuals with primary ciliary dyskinesia (PCD), a rare, ciliopathic, autosomal recessive genetic disorder affects the movement of cilia in the lining of the respiratory tract. Individuals with PCD suffer from reduced mucus clearance from the lungs, and susceptibility to chronic recurrent respiratory infections, as is the case with CF. By comparing monocytic- and epithelial- driven inflammation in CF and PCD, one is able to distinguish between inflammation due to recurrent infection, as is the case with both CF and NCFB, and inflammation that is downstream of CFTR/ENaC-mediated ionic disturbances, specific to CF.

The SAID patient cohort is composed of an array of systemic autoinflammatory diseases that are defined by an innate immune driven inflammation. The variety of autoinflammatory disorders described in this manuscript demonstrates the broad range of pathophysiology within this rare inflammatory disease spectrum. Here we demonstrate that the intrinsic ionic defect in cells and individuals with CF-associated mutations predisposes hyperactivation of the NLRP3 inflammasome, leading to inappropriate and destructive innate immune driven inflammation, as found in autoinflammation.”

6) In line with comment #2, it is clear that there are some outliers in the SAID group in Figure 3A and B. Are these the same patients? What was their underlying disorder? This information should be included at least in the legend to Figure 3.

The outliers in the SAID group in Figure 3A and 3B are the same patients. The highest outlier corresponds to HIDS 1 patient (shown in the updated Supplementary file 1) and the second highest outlier corresponds to A20 (TNFIP3) haploinsuficiency. This information has been included in the legend for Figure 3.

“Outliers in SAID group for IL-1β and IL-1Ra correspond to HIDS 1 and A20 deficiency”

7) Patients with SAID are often found to have ex-vivo IL-1beta production in the absence of an inflammatory stimulus. The question is whether this is the case in CF. It is a pity that in Figure C and D unstimulated cultures are not depicted. Are these measurements available? The blank measurements in Figure 2B are a bit of a concern as the IL-1beta concentrations are in the range of low dose LPS stimulation (and hence contamination).

To address this question we have included IL-18 and IL-1β measurements from unstimulated and stimulated (LPS or LPS/ATP) from all patient groups (SAID, HC, CF and NCFB). See the attached data in response to comment 2. These data shows that there are elevated levels of IL-18 relative to HC but not IL-1β in the unstimulated SAID samples. Additionally, *Ex-vivo* production of IL-18 and IL-1β is not increased in the unstimulated CF samples.

The authors agree that the unstimulated levels of IL-1β are higher than expected and may indeed be assay artefact due to high background levels that is often the case with difficult to measure cytokines, such as IL-1β. However, the authors wish to emphasise that the key conclusion of these data is that there is no difference between the HC and CF cohorts in terms of NLRC4, Pyrin or AIM2 inflammasome activation, regardless of the precise cytokine concentration observed.

8) In some experiments, all monocytes (CF, SAID and NCFB) are studied. In others, only monocytes from CF. The comparison with SAID monocytes is especially interesting in the Na/K experiments and for example the IFNgamma experiments. Why are they missing there?

As the reviewers will appreciate, the availability of such rare samples is extremely limited. In light of this, we prioritised experiments to ensure the comparison between the innate inflammation that governs both SAID and CF diseases was complete as possible. Unfortunately, the trade-off was insufficient sample to perform the two experiments that the reviewers highlight here. However, we would be confident in hypothesising that the ionic fluxes of SAID patients would be similar to that of the HC cohort and the IFNγ measurements would match that of CF, if not with a more pronounced IFN signature. These are experiments that we plan to perform in the future as further SAID patient samples become available for follow-up projects.

9) IL-1beta measurements in serum are most often below or close to the detection level of the commercial ELISA assays. The authors should inform us about the level of detection with their use of the assay. This also holds for the other ELISAs used.

The detection range of the IL-1β assay used in this study was 31.2-2000 pg/ml with assay sensitivity <31.2 pg/ml. The few values that were below the limit of the ELISAs used were extrapolated using the standard curve. The authors acknowledge that IL-1β is challenging to measure in the serum, due to its poor bioavailability and stability. However, we are confident that the data are accurate and reflect the extent to which the inflammasome pathway is active within these patient cohorts. In addition, IL-18, IL-1Ra, caspase-1 and ASC complement the IL-1β measurements, supporting systemic inflammasome activation in CF and SAID cohorts. We have included a resource table as part of the methods so that each ELISA kit used can be identified (https://www.thermofisher.com/elisa/product/IL-1-β-Human-Matched-Antibody-Pair/CHC1213).

10) In line with the previous comment: the serum cytokine concentrations measured are extremely high, even for healthy controls. Please check whether the assays were performed correctly, and calculations done right. Usually concentrations of circulating IL-6 are usually around 1 pg/ml (and not 50-200), IL-1beta below 1 pg/ml (here values ranging from 5–90pg/ml are reported). Are CRP values known for the participants?

We can confirm that the assays were performed as per the manufacturer’s instructions and the analyses have been further double checked. Circulating serum cytokines can vary between individuals, disease states and assay used, discounting experimental variations. Various ‘physiological’ ranges exist in the literature but ranges of 13-227pg/ml for IL-1β, 42-203pg/ml for TNF, 13-149pg/ml for IL-6 and 112-294pg/mL for IL-1Ra have been observed (Sekiyama et al., 1994) and support the data in this manuscript. CRP levels have now been included with the revised Supplementary file 1.

11) In subsection “Proinflammatory cytokines and ASC specks are elevated in CF sera, and are comparable to patients diagnosed with systemic autoinflammatory disease (SAID)” it suddenly becomes clear that the SAID patients were on Anakinra treatment. This should have been mentioned in the patient section.12) In connection with the previous comment: by blocking the IL-1 receptor, Anakinra will inhibit IL-1beta induced by IL-1(β and α). In general, this effect is rather limited if one looks at circulating IL-1beta. Of course, secondary cytokines like IL-6 and IL-8 will be lower. Thus, the sentence “All SAID patients were on active recombinant IL-1Ra (anakinraR) therapy, which will have reduced serum IL-1b levels.” should be rephrased.

We have included the following in the revised subsection “Clinical characteristics of patients”:

“All SAID patients were on Anakinra treatment, when blood samples were obtained”.

13) Subsection “Proinflammatory cytokines and ASC specks are elevated in CF sera, and are comparable to patients diagnosed with systemic autoinflammatory disease (SAID)”: how do the authors know that they detect endogenous IL-1Ra and not Anakinra?

Anakinra (17.3 KD) differs from the sequence of IL-1Ra (23-25 KD) by one methionine at the Nterminus. Anakinra is also not glycosylated. Whether these factors affect the ELISA assay’s ability to detect anakinra is unknown to the authors and therefore agree this there is a possibility that some of the IL-1Ra serum levels in the SAID patient cohort displayed in Figure 3C may be anakinra ‘contamination’.

We have included a statement in the figure legend making this point clear.

“Of note, an undetermined amount of detected IL-1Ra is attributed to circulating Anakinra (recombinant IL-1Ra) specifically in the SAID cohort.”

14) It is hard to believe that the genotypes of the SAID patients (with the exception of the Schnitzler patient) are not known (Supplementary Table).

The authors regret to have not included this information in the original paper and have now included the known genotypes for the SAID patients in the revised Supplementary file 1.

15) In the PBMC stimulation experiments (in the Materials and methods as well as in the figure legends) details are missing. How long were the cells stimulated, how many cells, and how much stimulus?

Two million PBMCs were used for this experiment. This is detailed in subsection “PBMC and monocyte isolation”. The following details have been included in Figure 2—figure supplement 1C, D legend where PBMCs were used.

“PBMCs were unstimulated or stimulated with LPS (10ng/ml, 4 hours) or LPS (10ng/ml, 4 hours) and ATP (5mM) for the final 30 minutes.”

16) The ATP preparation is missing in the Materials and methods section, how was ATP prepared for these experiments? This is of great importance as described by Stoffels et al., 2015. Please add methods.

The following details on ATP preparation have been included (subsection “Cell stimulations”):

“ATP was dissolved in pre-warmed at 37°C medium (100 mM stock) and immediately (~2 minutes) added to the cells”.

17) Although the discussion on the disparity between IL-1 and IL-18 production is interesting (Discussion section), the reasons why the cell lines do not show IL-1beta production should also be discussed.

We feel we have discussed this disparity between the cell lines adequately in the Discussion section. We have also moved the paragraph referred to in the comment to follow directly on from this discussion as it fits better here.

18) In the same vein, there should be some discussion of the responses of the different mutation HBEC cell lines. Generally, there seems to be the most significant effects with the IB3-1 line but this does not correlate with the level of b-ENaC observed by Western blotting (Figure 4G).

We have expanded on this with the following addition (Discussion section):

“Cystic fibrosis is a monogenic disease, yet consists of varying degrees of pathology depending on the precise CFTR mutation and the extent to which the encoded CFTR protein is subsequently transcribed, translated, folded within the ER, expressed on the plasma membrane, its stability on the surface and its ability to conduct chloride and bicarbonate. There are well documented examples of different classes of CFTR mutations that fail at each one of the above stages of CFTR expression and function. The IB3-1 (ΔF508/W1282X) cell line is associated with the greatest amounts of inflammation. The W1282X mutation is associated with severe disease clinically, that can be attributed to little to no protein expression. Inflammation in the IB3-1 cell line in vitro does not correlate with ENaC protein expression. The underlying mechanism of an ENaC/NLRP3 axis driving inflammation in cells with CFassociated mutations may not be common to all mutation classes. Notably, all of the cells (monocytes and epithelia) had at least one ΔF508 allele. Whether the loss of control of ENaC expression and function observed in ΔF508 CFTR expressing cells is unique to this more common mutation will require further study. It is important to note that inhibition of ENaC alleviated NLRP3 inflammasome-mediated inflammation in all cell lines (Figure 5 E-F)”.

19) The Discussion section again starts with an introduction. Instead it should start mentioning the main findings of the paper, and deal with the aims that were set at the end of the Introduction. So at least skip the first paragraph and rephrase the sentences that follow.

We have rephrased the discussion appropriately with the following (Discussion section):

“Here we have described our findings that IL-1-type cytokines were elevated in both monocytes and serum of patients with CF, but two of the major inflammasome-independent cytokines, TNF and IL-6, were not significantly elevated in these patients. In contrast to patients with SAID, these data suggest that the proinflammatory cytokine response in CF has a predominant NLRP3 inflammasome constitution. Furthermore, on NLRP3-inflammasome activation, IL-18 secretion was upregulated in the CF-associated mutant cell lines, IB3-1 (p<0.0001) and CuFi-1 (p<0.0001) relative to the BEAS-2B control, and these levels were reduced by treatment with small molecule inhibition of NLRP3-inflammasome signalling, thereby confirming the NLRP3-inflammasome as a major source of the elevated IL-18 inflammatory cytokine in these cells. Perhaps the most notable feature is the uncovering of a molecular link between enhanced ENaC-dependent Na^+^ influx, observed in cells with CF-associated mutations, and the exacerbated NLRP3 inflammasome activation. By pretreating these cells with CF-associated mutations with small molecule inhibitors of ENaC, the exaggerated IL-1-type cytokine response in vitro was diminished (Figure 7)”.

20) The authors are very prudent discussing the potential therapeutic consequences (Discussion section). Anakinra would be worth trying, as it does not meet with major infectious complications (at least much less so than Canakinumab). In fact, it has been given successfully to patients with Chronic granulomatous disease (who also suffer from excess IL-1beta production, but at the same time have a serious immunodeficiency).

Anakinra may have a role in the management of CF. And we have previously used the drug to manage a complex case of athropathy in CF (unpublished). Despite significant improvements in joint symptoms there was no improvement in lung function. Therefore, modifying the aberrant IL-18 as well as IL-1b response may be necessary to downregulate autoinflammation in CF. A phase 2 trial using anakinra in CF is is progress (EudraCT Number: 2016-004786-80).

We have proposed this in the revised Discussion section:

“Anakinra is a viable therapeutic option for CF, particularly with its short half-life and daily treatment regime allowing a more controlled dosage during any inevitable infections. In fact, a phase IIa clinical trial to evaluate safety and efficacy of subcutanous administration of anakinra in patients with cystic fibrosis is in progress (EudraCT Number: 2016-004786-80).”

21) A scheme summarizing the mechanism and sequence of events would help the readers.

We have included the following schematic diagram and legend in the paper, as figure 7.

“Figure 7: A schematic diagram of the proposed excessive NLRP3 inflammasome activation observed in individuals and cells with CF-associated mutations. Without functional CFTR, inhibition of ENaC currents is diminished leading to increased intracellular Na^+^ levels. Dysregulation of ENaC-dependent Na^2+^ influx leads to increased K^+^ efflux (via unknown mechanism) and NLRP3 inflammasome activation, with subsequent release of IL-1β and IL-18. In the CF airway, K^+^ efflux is exacerbated upon K^+^ channel stimulation by endotoxins, DAMPs or PAMPs, leading to aberrant NLRP3 inflammasome activation and excessive IL-1β and IL-18 secretion. Blocking ENaC currents with S18 peptide restores Na^+^ and K^+^ levels which reduces NLRP3-mediated production of IL-1β and IL-18.”

22) As a suggestion: the authors might discuss why CF patients that are constantly exposed to Pseudomonas spp do not show endotoxin-tolerant monocytes.

We have included the following the revised Discussion section:

“There is evidence in the literature of endotoxin-tolerant monocytes from patients with CF, with reduced cytokine expression in response to repetitive endotoxin exposure [70-72]. However, we propose that peripheral blood monocytes, such as those used in this study, are not constantly exposed to common pathogenic antigens that exist within the lung microenvironment during infections. In fact, one may propose a scenario of trained immunity, whereby innate immune cells hyper-respond to the recurrent infections, with greater destructive consequences. We also suggest that endotoxins are not the only inflammatory stimulus for NLRP3 inflammasome priming that exist in the inflammatory milieu of the CF lung. Epithelial-derived cytokines, neutrophilic extracellular traps (NETs), necrotic cell death and subsequent DAMPs may all induce transcription of IL-1β /IL-18 and other inflammasome components, independent of endotoxins. A caveat to our study is that we have not tested the full array of DAMPs and PAMPs that exist in the CF lung that may trigger an intrinsic predisposition to NLRP3 inflammasome activation observed in this study”.